# Multiomic profiling of the acute stress response in the mouse hippocampus

Lukas M. von Ziegler[1,2,10], Amalia Floriou-Servou [1,2,10], Rebecca Waag [1,2], Rebecca R. Das Gupta [3,4], Oliver Sturman[1,2], Katharina Gapp [1,2], Christina A. Maat[1,2], Tobias Kockmann [5], Han-Yu Lin [1,2,6], Sian N. Duss [1,2], Mattia Privitera [1,2], Laura Hinte [7], Ferdinand von Meyenn [7], Hanns U. Zeilhofer [2,3,4], Pierre-Luc Germain [2,8,9] & Johannes Bohacek [1,2✉]

The acute stress response mobilizes energy to meet situational demands and re-establish homeostasis. However, the underlying molecular cascades are unclear. Here, we use a brief swim exposure to trigger an acute stress response in mice, which transiently increases anxiety, without leading to lasting maladaptive changes. Using multiomic profiling, such as proteomics, phospho-proteomics, bulk mRNA-, single-nuclei mRNA-, small RNA-, and TRAP-sequencing, we characterize the acute stress-induced molecular events in the mouse hippocampus over time. Our results show the complexity and specificity of the response to acute stress, highlighting both the widespread changes in protein phosphorylation and gene transcription, and tightly regulated protein translation. The observed molecular events resolve efficiently within four hours after initiation of stress. We include an interactive app to explore the data, providing a molecular resource that can help us understand how acute stress impacts brain function in response to stress.

[1] Laboratory of Molecular and Behavioral Neuroscience, Institute for Neuroscience, Department of Health Sciences and Technology, ETH Zurich, Zurich, Switzerland. [2] Neuroscience Center Zurich, ETH Zurich and University of Zurich, Zurich, Switzerland. [3] Institute of Pharmacology and Toxicology, University of Zurich, Zurich, Switzerland. [4] Institute of Pharmaceutical Sciences, ETH Zurich, Zurich, Switzerland. [5] Functional Genomics Center Zurich, ETH Zurich and University of Zurich, Zurich, Switzerland. [6] Institute of Pharmacology and Toxicology, University of Zurich-Vetsuisse, Zurich, Switzerland. [7] Laboratory of Nutrition and Metabolic Epigenetics, Institute of Food, Nutrition and Health, Department of Health Sciences and Technology, ETH Zurich, Zurich, Switzerland. [8] Computational Neurogenomics, Institute for Neuroscience, Department of Health Sciences and Technology, ETH Zürich, Zurich, Switzerland. [9] Laboratory of Statistical Bioinformatics, Department for Molecular Life Sciences, University of Zürich, Zurich, Switzerland. [10] These authors contributed equally: Lukas M. von Ziegler, Amalia Floriou–Servou. ✉email: johannes.bohacek@hest.ethz.ch

The acute stress (AS) response enhances the chance of survival by mobilizing the organism's energy resources to meet situational demands[1–3]. When the stressor has subsided, successful stress management requires the efficient termination of the stress response to avoid stress-related wear and tear[4–6]. Notably, the majority of individuals can successfully cope with stressful challenges and maintain mental health even in the face of severe stressors[7–9]. However, when AS becomes too intense, or occurs repeatedly (i.e. chronic stress), it can overwhelm the "healthy stress response" and give rise to neuropsychiatric diseases such as post-traumatic stress disorder, anxiety and depression[10–12]. To understand the underlying mechanisms, many studies have assessed how chronic stressors impact the genome-wide molecular landscape in the brain, particularly in the hippocampus (HC)[13–21]. However, much less is known about the molecular changes triggered by AS, and how they unfold dynamically over time and across different molecular scales. Previously published studies have shown gene expression changes at individual time points after AS exposure[14,22–26]. Thus, we only have a fragmented picture of how AS affects the molecular machinery in the HC, and many key questions remain unanswered. First, how do stress-induced effects on transcription evolve over time and how long do they persist? Second, in which cell types do these changes occur? Third, which molecular changes occur upstream of transcription, at the level of protein phosphorylation and transcription factor activity? Fourth, which molecular changes occur downstream of transcription, at the level of translation and protein regulation? Here, we use a multiomic approach to dissect the healthy stress response across molecular scales, by cataloguing the stress-induced changes at the phosphoproteomic, transcriptomic, translatomic and proteomic levels. We look across cell types and profile the molecular changes over time, to reveal how an organism mounts a peak stress response and then reestablishes homeostasis after a brief, intensely stressful experience.

## Results

**Stress-induced effects on behavior.** To induce AS, we chose the forced swim stress model, which strongly activates the sympathetic nervous system and the HPA axis[14,22,23,27]. We first confirmed previous work showing that mice favor avoidance over approach behaviors shortly after an AS exposure[28,29]. When tested 45 mins after an AS in the open field test (OFT), mice travelled less distance, spent less time in the center, and performed fewer supported and unsupported rears (Supplementary Fig. 1A/B). However, when tested 2, 4 or 24 hours after the initiation of stress, these changes largely disappeared. Stress-exposed mice were behaviorally similar to unstressed control mice in the OFT (except regarding the number of supported rears) (Supplementary Fig. 1A/C) and indistinguishable from controls in the elevated plus maze (EPM) (Supplementary Fig. 1A/D). This swift and almost complete recovery observed within 24 hours indicates that–despite a strong initial response–mice cope successfully with this potentially life-threatening experience and show no long-lasting alterations in anxiety-related behavior. Therefore, the acute swim stress model appears to be suitable for studying successful, "healthy" stress coping mechanisms and the underlying molecular changes in the brain.

**Acute stress rapidly and transiently changes the phosphoproteome.** Some of the earliest intracellular molecular changes triggered by stress occur at the level of protein phosphorylation[30–32]. Changes in protein phosphorylation are highly dynamic, and they are typically not sustained in the face of

protein turnover. However, previous work has shown that after very intense stressors, some phosphorylation changes may persist for one hour[33] or even a day[34,35]. Thus, we first assessed protein phosphorylation in an unbiased way. Mice were euthanized either directly after the 6 min acute swim stress exposure, or at 15, 30 or 45 mins after the initiation of stress (Fig. 1A). Mice in the control group remained in their home-cage. The early time points were chosen based on previous literature[30,33], the later time points were determined empirically. Because the dorsal and ventral hippocampus (dHC and vHC) are engaged in different brain circuitries and are molecularly very distinct[22,36], we dissected them separately and performed label-free quantitative phosphorylation analysis by liquid chromatography-tandem mass spectrometry (LC-MS/MS). We used 5–8 mice per group (see Fig. 1A), and we treated each mouse as an independent biological sample. Overall, we were able to quantify 16,302 distinct modified peptide sequences in our phospho-enriched samples, of which 10132 (62%) were phosphopeptides. For all analyses we used FDR adjusted $p$ values $< 0.05$ to determine significance. Immediately after stress, we detected 847 significantly altered modified peptides (794 phosphopeptides; ~93%) in the dHC, and 269 in the vHC (253 phosphopeptides; 94%) (Fig. 1B,C). 15 mins after stress, this number dropped to 206 modified peptides in the dHC (188 phosphopeptides; 91%), and 94 modified peptides in the vHC (86 phosphopeptides; 91%). At the 30 and 45 mins time points, we could not detect significant changes anymore, indicating that protein phosphorylation changes after AS occur rapidly and seem to be tightly regulated and short-lived in the hippocampus. Significantly altered peptides were strongly enriched for phosphopeptides (fisher exact test; six min dHC: $p = 1.17$ e-15, odds ratio $= 1.5$, 6 min vHC: $p = 3.4$ e-6, odds ratio $= 1.51$), and phosphopeptides demonstrated stronger fold-changes compared to non-phosphorylated peptides (Supplementary Fig. 2C). To ensure that the observed phosphorylation changes are not due to overall changes in the proteome (protein abundance), we performed a quantitative proteomic measurement of reference samples (not enriched for phosphopeptides), which revealed no significant changes 6 mins after stress (Supplementary Fig. 2B). Importantly, the protein phosphorylation changes were correlated between the 6 and 15 min time points (dHC: $R^2 = 0.35$, vHC $R^2 = 0.39$), as well as between the 15 and 30 min time points (dHC: $R^2 = 0.35$, vHC $R^2 = 0.55$), suggesting that most of the phosphorylation events belong to one big wave of phosphorylation that unfolds over time (Supplementary Fig. 3C). In line with known molecular differences between dHC and vHC[22,37–39], we observed baseline differences in protein phosphorylation between the dHC and vHC (Supplementary Fig. 2D). Despite these baseline differences, there was a strong correlation between the stress-induced protein phosphorylation changes in the dHC and vHC, at both the 6 min ($R^2 = 0.42$) and 15 min ($R^2 = 0.48$) time points (Supplementary Fig. 3B). This suggests that even though we observed more significant phosphorylation changes in the dHC, similar and robust phosphorylation cascades are triggered throughout the longitudinal axis of the HC. Using $K$-means clustering, we grouped the significantly modified peptides into 5 distinct temporal profiles (Supplementary Fig. 3A) and assigned their corresponding proteins to gene ontology (GO) terms. Different statistical methods did not reveal a consensus on the best number of centers; thus, a manual approach was used, which may be subject to bias (see methods). Most modified proteins were related to dendritic morphology and development, calcium signaling and synaptic function, AMPA-receptor regulation and neurotransmitter release. This indicates stress-induced activation of neuronal activity in the HC. Across all stress-responsive phosphoproteins we found enrichment for a number of pathways[40], including calcium signaling

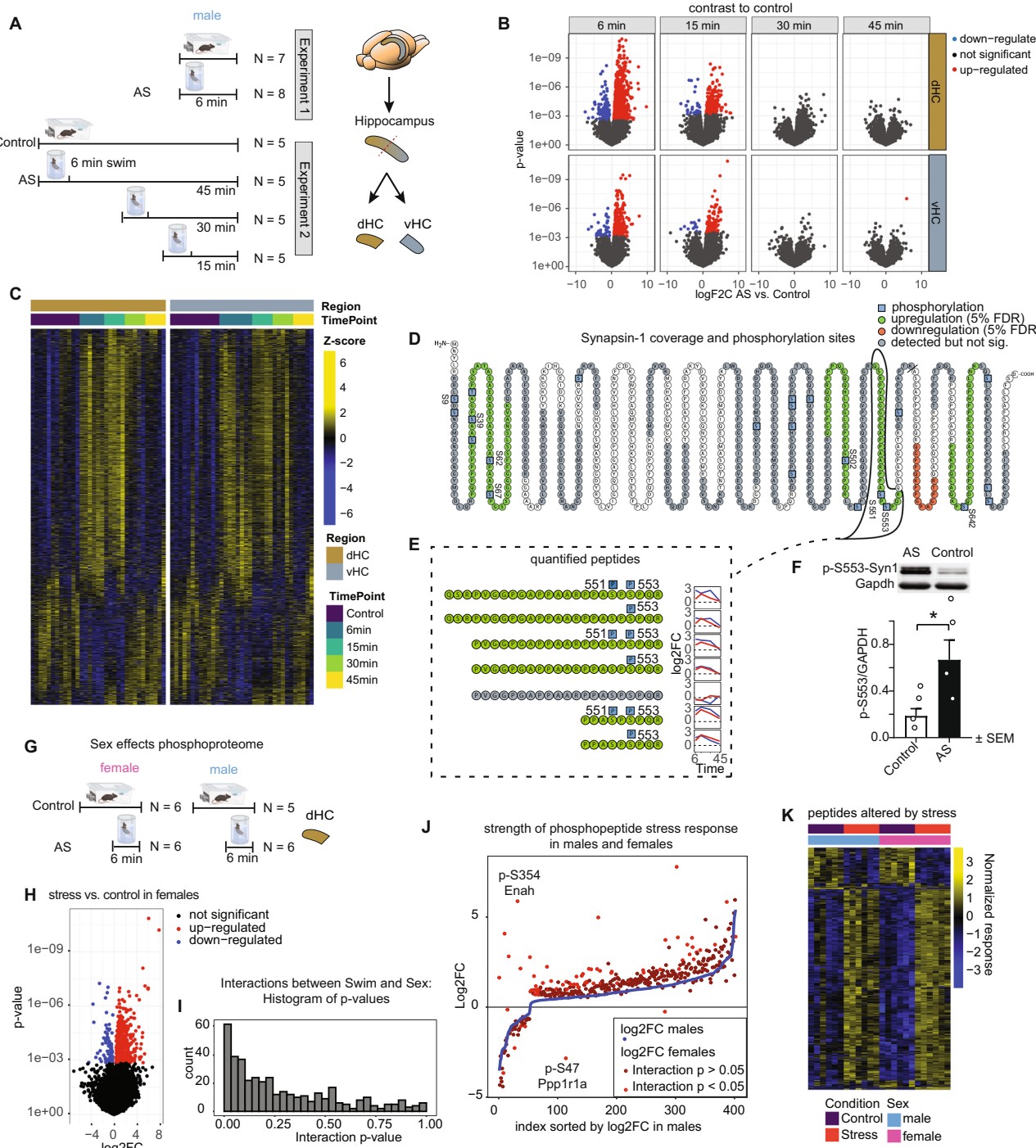

**Fig. 1 Rapid, short-lived effects of acute stress (AS) on the phosphoproteome of the dorsal hippocampus (dHC) and ventral hippocampus (vHC).**
**A** Experimental design and tissue collection approach. **B** Volcano plots showing the estimated log2 fold changes and statistical test results for modified peptides immediately (Exp. 1) or shortly (Exp. 2) after exposure to AS. Red and blue dots represent significant changes within 5% false discovery rate (FDR) relative to non-stressed controls. **C** Heatmaps showing the abundance of significantly modified peptides across all samples. **D** Amino acid sequence of Synapsin 1, illustrating overall coverage (grey, green and red), phosphopeptides significantly upregulated (green) or downregulated (red) at any time point after AS and detected phosphosites (blue, probability of post-translational modification >0.75). **E** Quantification of all detected Synapsin 1 modified peptides spanning sites pS551 and pS553, and their temporal profiles (blue = dHC, red = vHC). **F** Validation of the upregulation of the phosphorylated site SYN1-pS553 with Western blot in the dHC immediately after stress (Effect size ~300%, t(6) = 2.40, p = 0.037, one-tailed unpaired t test, N = 4 mice/ group). Data represent mean ± SEM. Source data are provided as a Source Data file (**G**) Experimental design of tests for phosphoproteomic sex differences. **H** Volcano plots showing the estimated log2 fold changes and statistical test results for modified peptides immediately after stress exposure in female mice. Red: significant upregulation (FDR adj. p < 0.05), blue: significant downregulation (FDR adj. p < 0.05) (**I**) Histogram of interaction p values of all stress or sex responsive phosphopeptides. **J** AS vs ctrl log2FCs in males (blue) and females (red) sorted by stress response strength in males. Phosphopeptides with significant AS:Sex interaction are labeled in bright red. **K** Stress responsive phosphopeptides in male and female dHC immediately after stress. *p < 0.05.

(Supplementary Fig. 2A), a pathway well-known to react to neuronal activation[41]. A domain enrichment analysis revealed that significantly altered phosphopeptides are overrepresented in two InterPro[42] protein domains, the Protein Kinase C and SAPAP (also known as Dlgap) family of proteins (Supplementary Fig. 2E), the latter is associated with synaptic function and is implicated in multiple neuropsychiatric diseases[43,44]. As the experiments reported thus far were performed in male mice, and because there have been reports of sex-differences in the stress response[45,46], we conducted an experiment to directly compare the stress-induced phosphorylation changes in males and females (Fig. 1G). We collected the dHC immediately after stress exposure and observed a strong effect of stress in females (Fig. 1H), revealing that most of the peptides that are regulated by acute stress in males are also regulated similarly in females (Fig. 1K). However, a two-way analysis revealed a significant interaction between stress and sex (Fig. 1I). Follow-up analyses revealed that only a small number of peptides showed different stress response patterns between sexes, but females generally responded more strongly than males (Fig. 1J).

Phospho-proteomics not only provides a detailed map of the complex and widespread phosphorylation changes detected across proteins, but it also reveals unique details about the phosphorylation changes within individual proteins. To illustrate this, we focused on Synapsin 1 (SYN1), a protein with well-characterized phosphorylation sites downstream of MAPK/ERK-CREB signaling[47–49]. Using a non site-specific antibody, SYN1 has been shown to be phosphorylated at various time points after AS in the HC[34], while another study reported increased S9 phosphorylation in the prefrontal cortex up to 24 h after stress[35]. By interrogating the whole SYN1 sequence, we detected both upregulated and downregulated phosphopeptides immediately after AS (Fig. 1D), with most detected peptides being phosphorylated at serine 553 upon stress (p-S553) (Fig. 1E). Indeed, western blot analysis validated a 3-fold increase in the levels of p-S553-SYN1 (Fig. 1F) in the dHC immediately after stress. Other significantly upregulated phosphopeptides contained the phosphosites S39, S62, S67, S502, S551, S642. Phosphorylation at S9 was also increased, but did not pass our FDR cutoff (logFC 6 min vHC 1.32, $p = 0.005$, adj. $p = 0.16$). Most of these changes in phosphorylation were still detectable at 15 mins, but diminished at later time points. Together, these data reveal the stunning complexity of protein phosphorylation within single proteins, the breadth of the phospho-proteomic changes triggered by AS, and the transient nature of the phospho-response as the animal successfully copes with an acute stressor.

**Stress-induced transcriptomic changes.** Protein phosphorylation triggers second messenger cascades that lead to changes in gene expression. However, it remains unknown how the stress-induced transcriptomic changes evolve over time, and whether these events unfold similarly in the dHC and vHC. Therefore, we profiled the transcriptome of the dHC and vHC at 45, 90, 120, 180 or 240 mins after the initiation of stress using bulk RNA-sequencing (RNAseq, Fig. 2A). The selection of these time points was based on previous work on stress-induced transcriptional changes[23,50]. We used 7–8 mice per group, and we treated each mouse as an independent biological sample. We observed highly dynamic gene expression changes over time in response to AS. Using 5% FDR-corrected $p$ values, the highest number of gene expression changes occurred at 45 and 90 mins after stress, followed by a gradual decline in both the dHC and vHC (Fig. 2B, D). After 4 h, no significant changes were detected anymore, suggesting a tight regulation of gene expression over time. We observed a high correlation in the gene expression changes

between consecutive time points up to 3 h after stress, indicating that these changes are not random but rather evolve systematically over time (Supplementary Fig. 3E). Overall, we detected more stress-induced gene expression changes in the vHC than in the dHC, consistent with our previous work[22]. However, we found that significantly changed genes correlated well between the dHC and vHC at all early time points (45 min: $R^2 = 0.42$; 90 min: $R^2 = 0.43$; 2 h: $R^2 = 0.48$) indicating a similar pattern of gene expression changes along the longitudinal axis of the HC (Supplementary Fig. 3D).

To understand what is driving these transcriptomic changes, we turned to previous work that described three distinct waves of gene expression triggered by neuronal activity in vitro[51]: First a translation-independent wave of rapid primary response genes (PRGs) that depend on ERK/MAPK signaling, then a delayed wave of translation-independent genes (delayed PRGs), and finally a translation-dependent set of secondary response genes (SRGs). While sustained neuronal activation triggered all three waves in vitro, brief activation mainly triggered only the first[51]. In line with this, AS fully recapitulates the first wave of rapid PRGs, whilst affecting only a minority of the delayed PRGs or SRGs (Supplementary Fig. 4E). Thus, the rapid response to AS likely stems from increased neuronal activity. Importantly, however, a number of delayed PRGs were already upregulated at 45 mins after AS, although they were not yet activated in vitro 60 mins after activation, suggesting accelerated dynamics in vivo. To gain more insight into the temporal dynamics in our dataset we next clustered the stress-regulated genes based on their temporal transcriptional profiles (using K-means with manually selected number of clusters, see methods), and we characterized the clusters in terms of their top enriched GO biological processes (Fig. 2E). Genes whose expression peaked at the earliest time point and then rapidly decreased (profiles 3 and 16) were associated with ERK/MAPK-CREB activity and the regulation of transcription. In contrast, we identified a second wave of genes that peaked at 90 mins, which were associated with glucose homeostasis (profile 10) and metabolism (profile 25), while apoptotic programs (profile 7) were negatively regulated. Later time points did not reveal novel transcriptional programs, but rather seemed to represent a gradual return to baseline levels of transcription.

In parallel to the waves of increased transcription, an equally large number of genes rapidly decreased in expression after stress exposure. This could be the consequence of active RNA degradation, mediated for instance by micro-RNAs, or due to rerouting of the transcriptional machinery and normal RNA decay. To investigate the former, we performed small RNA sequencing of both hippocampal regions 45 mins after stress. While quality control was successful (Supplementary Figs. 5A, B), we found no significant change in miRNA expression, nor any significant miRNA target enrichment in the transcriptional signature (Supplementary Fig. 5C, D). To test the hypothesis that the rapid down-regulation was instead the result of normal mRNA decay, we plotted the fold changes for genes of different half-lives, as estimated in unperturbed mouse fibroblasts[52]. This revealed that genes with shorter half-lives were more strongly downregulated (Fig. 2C). In addition, downregulated genes have a median half-life of 4.5 hours in the dish, with an expected passive downregulation of 11% after 45 mins, which is consistent with the observed median downregulation of 12% across the same genes 45 mins after AS. Therefore, we speculate that the major transcriptional activity triggered by AS occurs at the expense of other genes, which decrease largely due to normal RNA decay.

Since we observed male-female differences in the phosphoproteome, we further compared the transcriptional response to acute stress in males and females (see Fig. 2F for experimental design).

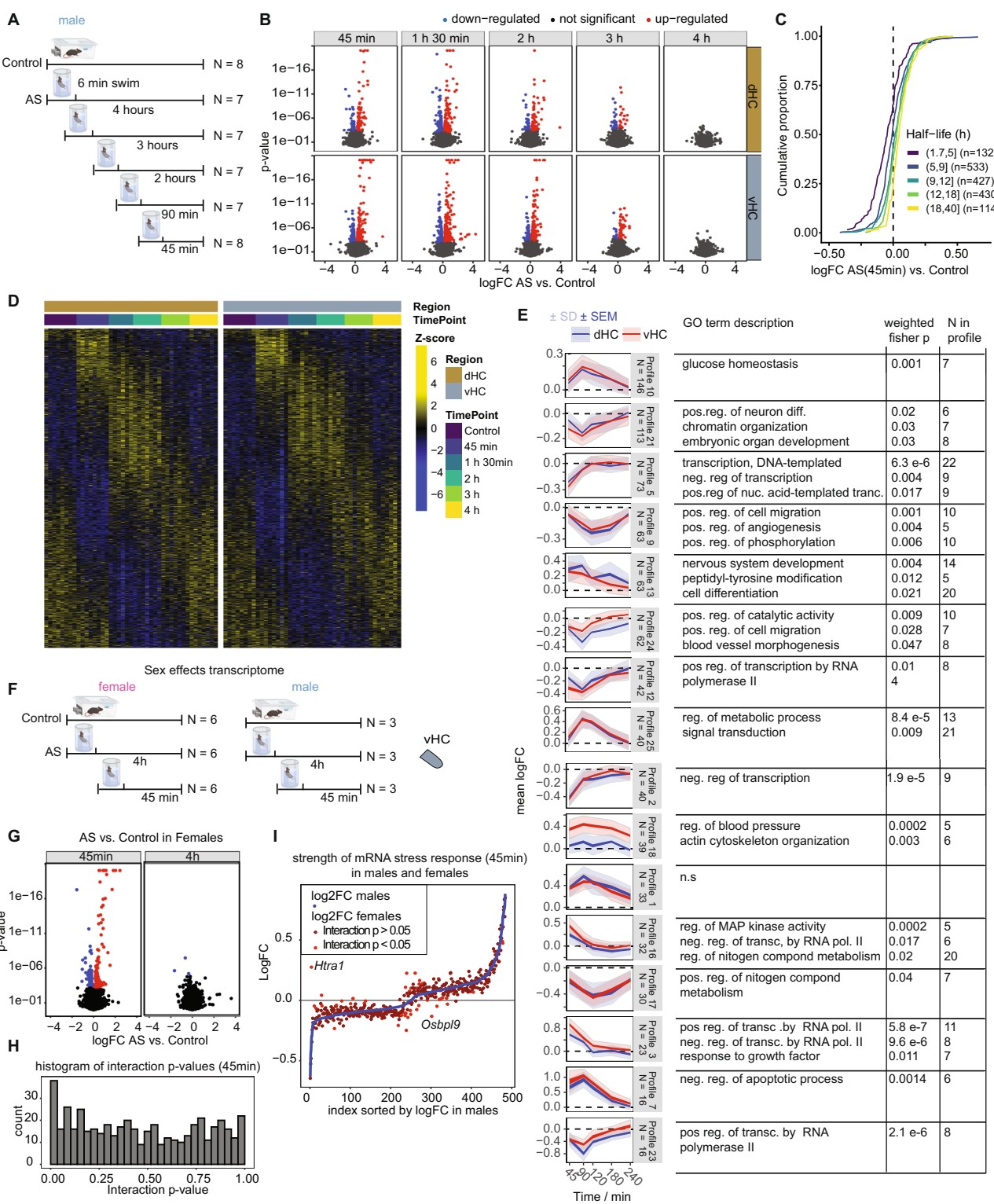

We faithfully reproduced the strong transcriptional response observed in males at the 45 min time point in females (Fig. 2G), and we found that both males and females showed a very similar response (Supplementary Fig. 3F). Combining male samples from the time-series experiment with samples from the sex-differences experiment allowed us to analyze an interactive sex*AS model with a high number of biological replicates (6 x 6 for females, 9x9 for males). Again, we found many stress-responsive genes at the 45 min time point (Supplementary Fig. 3G), but none at the 4 hour time point (Supplementary Fig. 3H). In addition, a number

of genes were differentially expressed between males and females across conditions (Supplementary Fig. 3I). We noted however, that all these genes were X- or Y-chromosomal genes, supporting the robustness of our approach and suggesting that most other stress-sensitive genes respond similarly in males and females. We observed that the interaction $p$ values of sex- or stress-responsive genes were slightly shifted towards 0 (Fig. 2H), indicating that there may be subtle sex differences in the transcriptional response to acute stress, which would require more targeted experiments to further validate. Plotting logFCs of males vs. females demonstrated

**Fig. 2 The effects of acute swim (AS) stress on the transcriptome of dorsal hippocampus (dHC) and ventral hippocampus (vHC) across time.**
**A** Experimental design (experiment in male mice). **B** Volcano plots showing the log fold change and statistical results of RNA transcripts at 5 different time points after exposure to AS. Red and blue values represent changes within 5% FDR. ($N = 7$–8 per group). Ceiling for $p$ values was set to 1E-20. **C** Cumulative fold change distributions of genes with different half-lives, consistent with the downregulation being the result of normal mRNA decay. **D** Heatmaps showing expression of all differentially-expressed genes in all samples. **E** $K$-means clustering of differentially expressed RNA transcripts to resolve temporal profiles, and their GO term description. $K$-means clustering was done with 25 centers, of which the ones with fewer than 15 transcripts were excluded. Clusters are ordered by decreasing number of genes. Blue indicates expression in the dHC, red in the vHC. Plots indicate both standard error of the mean (SEM) of logFC in bright color and standard deviation (SD) in faint color. n.s no significant enrichment. **F** Experimental design of tests for transcriptomic sex differences. **G** Volcano plots depicting statistical results of stress effects at 45 mins and 4 hours in the vHC transcriptome of females. Red: significant upregulation (FDR adj. $p < 0.05$), blue: significant down-regulation (FDR adj. $p < 0.05$) (**H**) Histogram of interaction $p$ values of all stress or sex responsive mRNAs. **I** AS vs ctrl logFCs in males (blue) and females (red) sorted by stress response strength in males. mRNAs with significant AS: Sex interaction are labeled in bright red.

that - in contrast to the phosphoproteome - the overall magnitude of the response was quite similar in both sexes (Fig. 2I). One notable exception was *Htra1*, which encodes HtrA Serine Peptidase 1, and which was suppressed by AS in the vHC in males but increased in females.

Going beyond sex-specific effects, we then addressed whether the transcriptional response to AS is similar in the left and right hemispheres, as lateralization effects had previously been reported[53–55]. A direct comparison of RNAseq data from left vs. right hemisphere in the same cohort of mice 45 mins after AS (Supplementary Fig. 4A) shows that the molecular response in both hemispheres is very similar (Supplementary Fig. 4B), with no significant interaction between hemisphere and treatment (Supplementary Fig. 4C).

**Active nuclear transcription and transcription factor activity after stress.** In the process of mRNA maturation, RNA is first transcribed in the nucleus as unprocessed pre-RNA, before being spliced relatively quickly and transported out of the nucleus for translation[56]. Therefore, we used reads from unprocessed transcripts as a proxy for actively ongoing nuclear transcription. By independently analyzing processed and unprocessed transcripts, we found that altered nuclear transcription mainly occurs early, at 45 mins after stress, and to a far lesser extent at the 90-min and 2-h time points (Fig. 3A). In contrast, processed mRNAs persist at later time points and resolve more slowly (Fig. 3A). A PCA analysis demonstrates that the unprocessed response is shorter in duration and relaxes much faster along the first principal component in contrast to the processed response (Fig. 3B). Also, within the profile clusters described above, unprocessed expression changes precede and predict expression of processed transcripts (Fig. 3C). The independent analysis of processed and unprocessed transcripts further enables more accurate estimates of transcription factor (TF) activity. Many TFs have complex modes of action (e.g. where they are activated through post-translational modifications, altered shuttling to the nucleus and/or binding to other cofactors), which cannot be inferred from their RNA levels. In addition, our phosphoproteomic data did not detect many TFs, preventing us from directly linking phosphorylated peptides to TF activity. We therefore opted to infer TF activity from the relative expression of their targets' unprocessed transcripts based on curated regulons[57]. We identified genes encoding TFs for which our data suggest increased activity mainly at the early 45 min time point (Fig. 3D and Supplementary Fig. 4D). One example is *Creb1*, which is activated through calcium signaling[58], a pathway for which we saw many of its components (i.e. signaling proteins such as CAMKs, PKC and ERK/MAPK) altered in our phospho-proteomics data (Supplementary Fig. 2A,E). To identify distinct transcriptional programs underlying the observed expression changes, we clustered differentially expressed genes based on both their temporal

expression profile and their inferred regulation by the TFs. This revealed transcriptional programs with significant enrichment for very distinct biological processes, which again include glutamatergic and synaptic networks (Fig. 3E). Overall, AS triggers active transcription very rapidly, and during the subsequent 4-hour period, waves of gene expression unfold before fully returning to baseline.

**Specific stress-induced transcriptomic changes in different cell types.** To test whether stress-induced transcriptional changes occur in specific cell types, we performed single nucleus RNA sequencing (snRNAseq) after AS. Because nuclear RNA was sequenced, we focused on the 45 min time point, when nuclear transcription is most active. We observed a total of 5292 QC-filtered nuclei in four independent hippocampal samples from two control mice and two stress-exposed mice. A combination of clustering and reference-based assignment produced distinct cell clusters linked to specific cell types or families (Fig. 4A), with a good segregation of known marker expression (Fig. 4B). We first wanted to see in which cell types differentially expressed transcripts from the bulk sequencing are most abundantly expressed. We found that overall, these genes were mainly expressed in neurons, astrocytes, oligodendrocytes, and vascular cells, but less so in microglia and oligodendrocyte precursor cells (OPC) (Fig. 4C). Most upregulated genes from bulk sequencing show the highest expression in astrocytes. To assess the extent to which snRNAseq results mirror bulk sequencing results in the response to AS, we then performed differential expression analysis in each subpopulation, and analyzed how well their log-fold-changes correlated to the (unprocessed) bulk response. Neurons and astrocytes showed the best correlation with bulk mRNA (Fig. 4D). However, since these correlations could be confounded by the relative abundance of the different cell types, we confirmed these results with an alternative assignment-based approach (see methods). In line with the correlations, we found that most expression changes (49%) were assigned to glial cells, whereas a smaller number of gene expression changes were assigned to neurons (24%) and vascular cells (18%) (Fig. 4E). In agreement with this, most of the TFs predicted to have differential activity after stress - based on the bulk RNAseq data—were preferentially expressed in non-neuronal cell types, although some showed a clear neuronal enrichment (Fig. 3D, right panel). The cell-type-wise differential expression analysis based on the snRNAseq data additionally revealed many genes that showed strong, cell-type specific regulation upon stress (Fig. 5A). *Apold1*, a known endothelial gene[59] that is strongly upregulated by AS[22,23], indeed showed a strong stress-induced increase only in vascular cells. In contrast, genes such as *Dio2* and *Map3k19* seem to be selectively activated in astrocytes. Gene set enrichment analysis (GSEA) further revealed extensive differences in the cell types' response to stress (Fig. 5B). Furthermore, we found important differences in

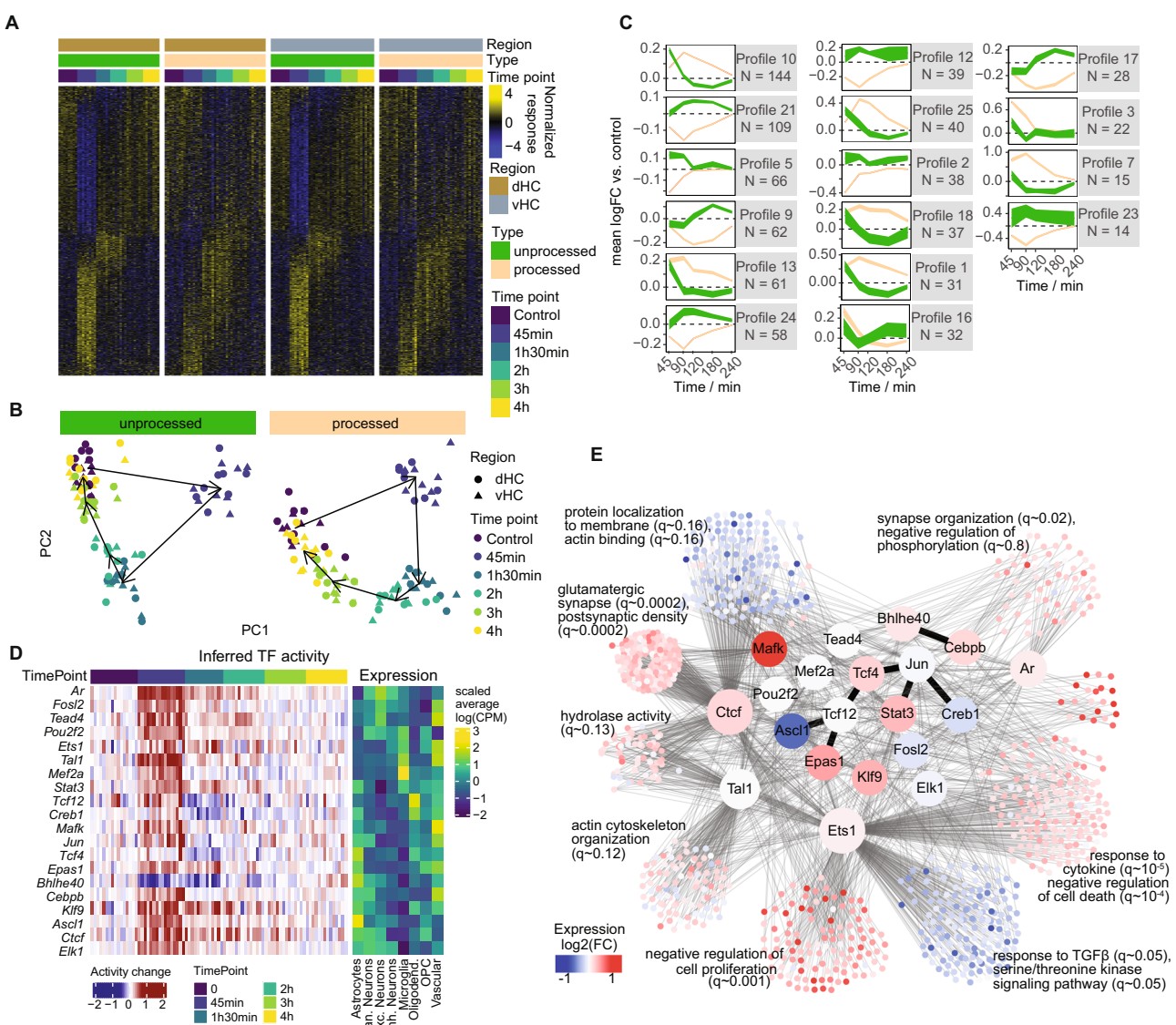

**Fig. 3 Active nuclear transcription after acute stress. A** Heatmap showing unprocessed (actively transcribed) vs processed transcripts of differentially expressed genes. **B** PCA plot of unprocessed and processed transcripts across differentially expressed genes. **C** Profiles from Fig. 2E resolved by unprocessed (green) and processed (brown) transcripts. **D** Relative activity of the most significant transcription factors (TF), inferred from the expression of their targets. The left panel shows inferred relative TF activity across bulk samples, while the right panel shows the relative average expression of the factors across cell types in the snRNAseq. **E** Network representation of the curated interactions between differentially expressed genes and the TFs found to be differentially active. Colours indicate the direction of differential expression at 45 mins. Thin lines represent transcriptional interactions, while dark, thick lines represent protein-protein interactions between the factors. For each cluster, the top GO enrichments are shown. CPM counts per million.

the upregulation of known activity-dependent genes, which could not be explained by their baseline expression in different cell types (Fig. 5C). Overall, these results highlight that - beyond neurons - glial cells and vascular cells contribute strongly to the overall transcriptional response to stress, and that this response is also highly cell-type specific.

**Stress-induced effects on the translatome**. Our transcriptomic analyses have revealed widespread and profound changes in gene expression across time and in different cell types. To determine how many of these transcripts will then become actively translated, we measured actively translated RNA using translating ribosome affinity purification (TRAP) followed by RNA sequencing (TRAPseq)[60,61] (Fig. 6A). To target actively translated RNA in all brain cells, we crossed CMV-Cre mice[62] with a floxed-nuTRAP mouse line[63]. CMV-nuTRAP mice ubiquitously express

a ribosomal green fluorescent protein (eGFP) tag fused with the ribosomal subunit L10a (Fig. 6B) and an mCherry tag fused with the nuclear pore protein RanGAP1. This experiment was only performed in females, because the breeding process generated few double-transgenic males. To target actively translated RNA specifically in excitatory neurons, we crossed CaMKIIa-Cre mice with floxed-nuTRAP mice (Fig. 6C). To target inhibitory neurons, we used the vGAT::bacTRAP mouse line[64], expressing the same eGFP ribosomal tag under the vesicular GABA transporter (vGAT, Slc32a1) promoter (Fig. 6D). CaMKIIa-NuTRAP and vGAT::bacTRAP experiments were conducted in males.

To validate the TRAP method, we performed RT-PCR to compare the expression of mRNAs and long non-coding RNAs in CMV-nuTRAP pre and post immunopurification (IP) fractions. We found a significant depletion of lncRNAs, whereas mRNAs were not changed (Fig. 6E). Further, we confirmed expression of

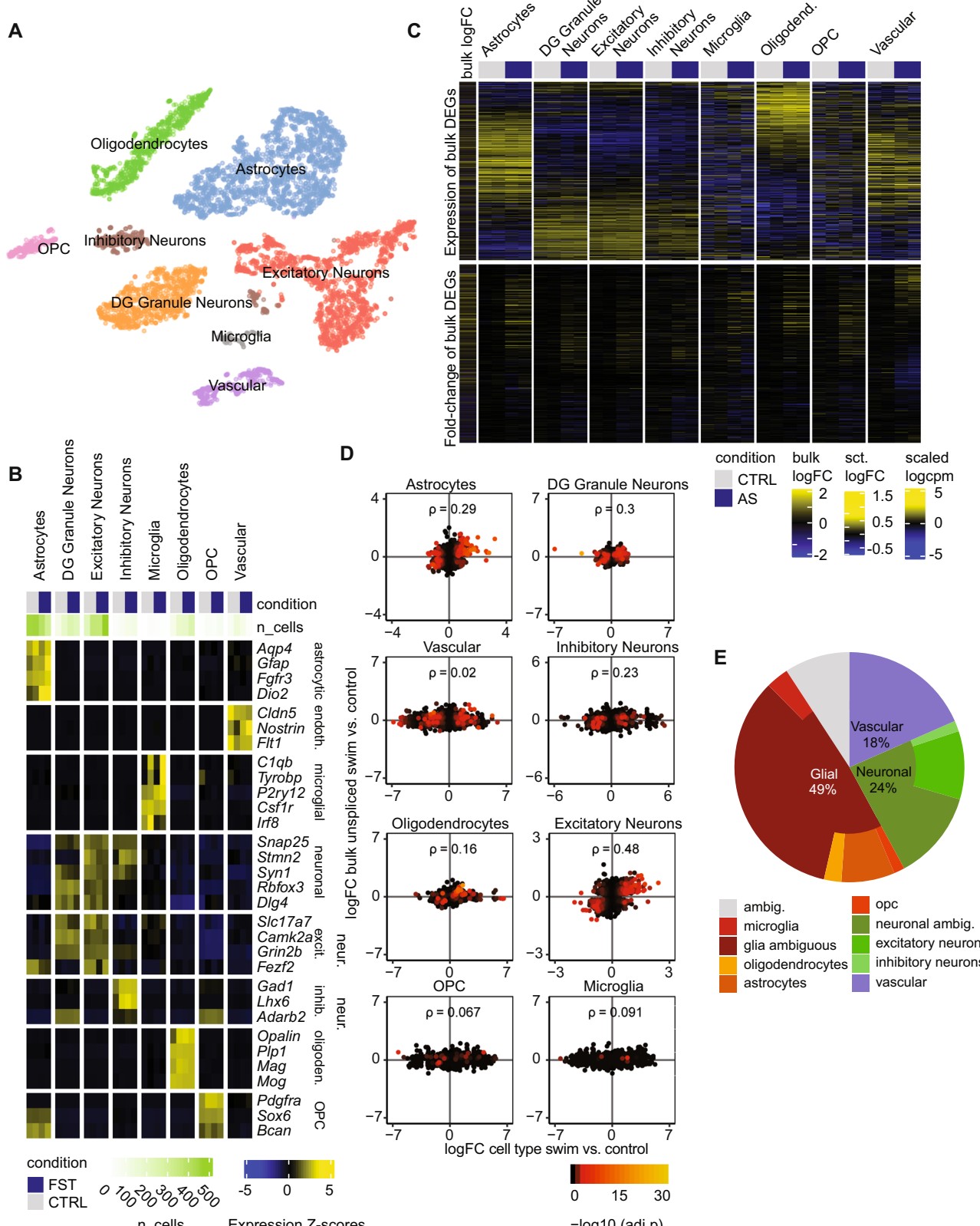

**Fig. 4 Single nucleus RNA sequencing 45 minute after acute stress (AS). A** t-SNE plot of single-nuclei transcriptomes labeled by cell-type. **B** Expression of known markers across the final snRNAseq clusters. Cell-level counts were aggregated by sample and cluster and normalized at the pseudo-bulk level. **C** Top: Heatmap indicating overall expression of stress responsive genes (from bulk sequencing) in logcpm in snRNA cell types. Bottom: Heatmap indicating logFC after AS of these genes in the snRNA data. **D** Correlations between logFCs of DEGs in bulk sequencing (45 min unprocessed) and logFC of snRNAseq data within cell types (ρ = Spearman's rank correlation coefficient). Color indicates adjusted *p* value in snRNAseq data. **E** Decomposition analysis highlights which cell types best explain DEGs from the bulk sequencing in the snRNA data. Experiments performed in male mice.

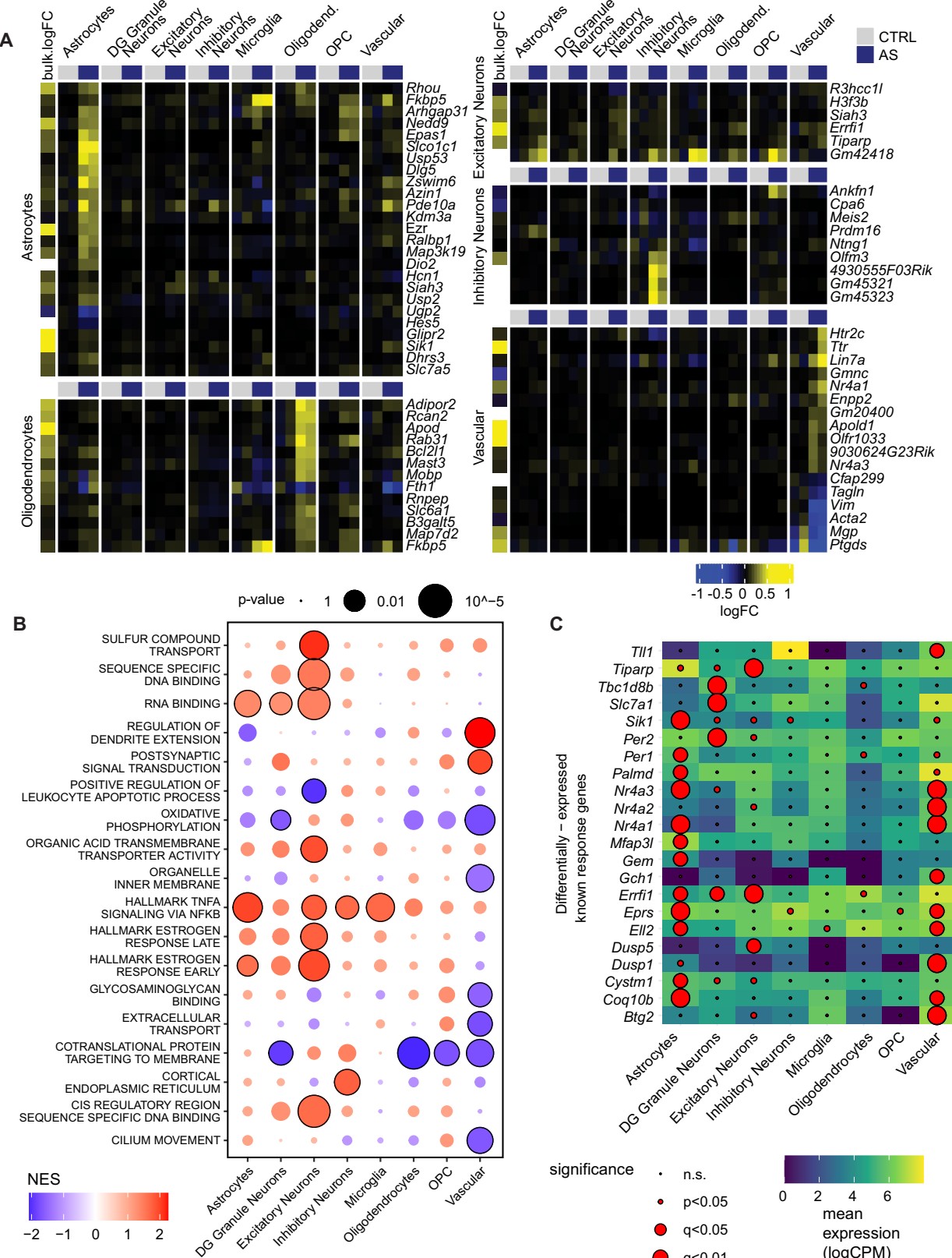

**Fig. 5 Differential expression analysis of snRNA data 45 mins after acute stress (AS). A** Top gene candidates from the differential expression analysis in snRNAseq. Each heatmap shows the top candidates of a selected cell-type and their expression in single-nuclei aggregated by cell type. At the left of each heatmap is the logFC in unprocessed bulk samples 45 mins after AS (white indicates insufficient data). **B** Gene set enrichment analysis (GSEA) on the stress signature across cell populations. To avoid redundant terms, the union of top significant terms across cell types were clustered based on gene memberships and the most significant term per cluster was retained. **C** known activity-induced response genes (from[50], with their significance vs mean expression levels in different cell populations. Experiments performed in male mice. NES normalized enrichment score, CPM counts per million.

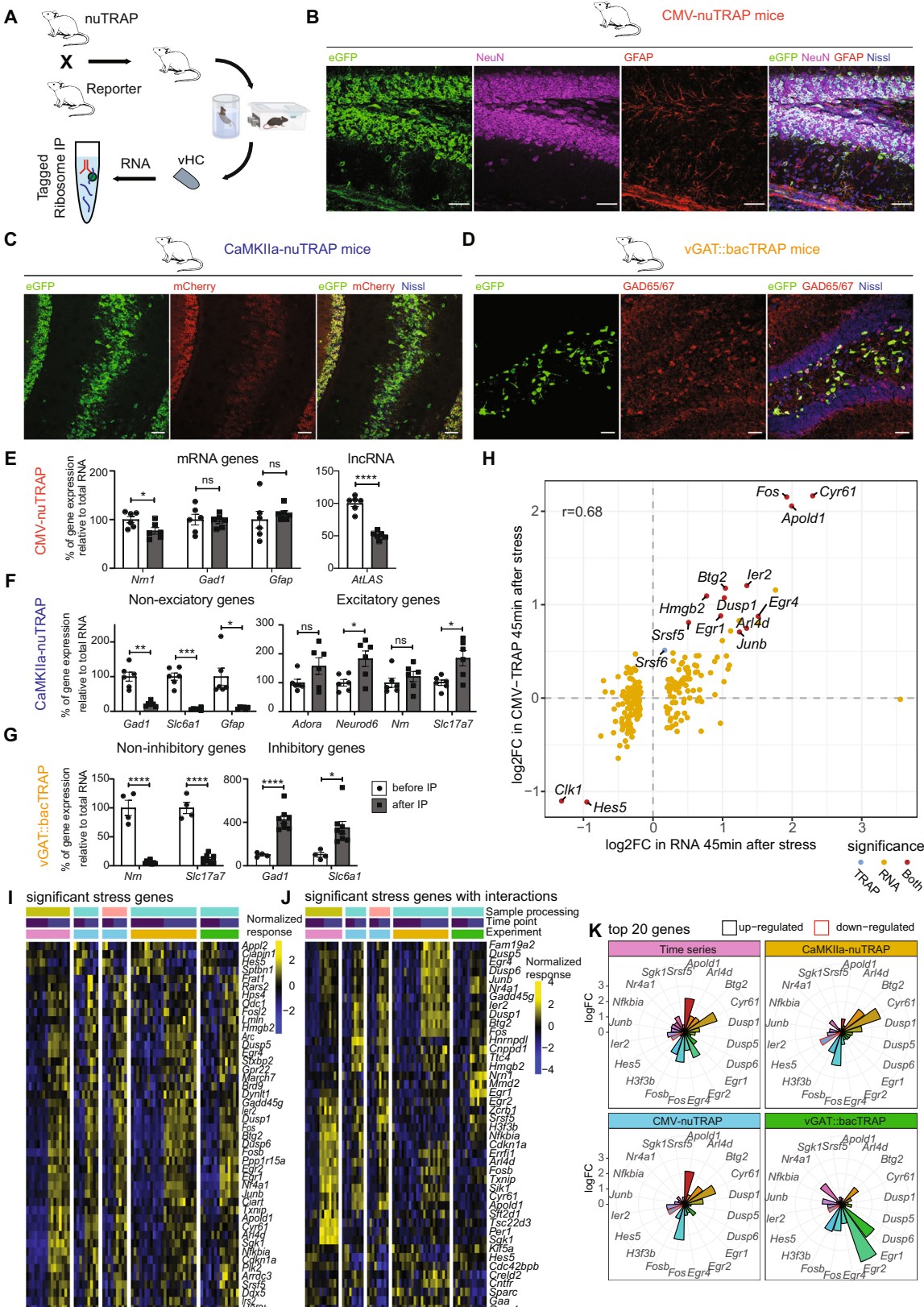

inhibitory and excitatory genes in the CaMKIIa-nuTRAP and vGAT::bacTRAP samples before IP (total RNA) and afterwards (bound RNA). As expected, inhibitory and glial genes were depleted, and excitatory genes were largely enriched in the bound RNA fraction of CaMKIIa-nuTRAP HC samples (Fig. 6F). Conversely, excitatory genes were depleted and inhibitory genes

were enriched in the bound RNA fraction of vGAT::bacTRAP tissue (Fig. 6G). However, the enrichment of excitatory genes seems to be less efficient in CaMKIIa-nuTRAP samples. This is likely due to several reasons: (1) excitatory neurons are the majority of neurons in HC, and hence it is harder to show enrichment of this abundant cell type. (2) These genes might not

**Fig. 6 Acute stress (AS) effects on the bulk translatome, and the translatome of excitatory and inhibitory neurons of the ventral hippocampus (vHC).**
**A** Schematic of TRAP. **B-D** Microscopy images showing co-localization of eGFP with NeuN and GFAP in the hilus and DG of CMV-nuTRAP mice (**B**), eGFP and mCherry in the ventral hilus, CA3 and DG of CaMKIIa-nuTRAP mice (**C**), and eGFP with GAD65/67 in the dorsal hilus and DG of vGAT::bacTRAP mice (**D**). Scale bars 50 μm. Experiments performed 1-2 times with similar results, images representative of <4 mice/line. **E–G** RT-qPCR validation of TRAP enrichment. Individual values represent dHC and vHC samples from 3–4 mice/line. Data represent mean ± SEM, asterisks indicate $p$ values generated by independent unpaired t-tests. Source data are provided as a Source Data file (**E**) Genes primarily expressed in excitatory neurons, inhibitory neurons or glial cells are present both before (total RNA) and after (bound RNA) the IP, while the lncRNA *AtLAS* is reduced in the bound RNA of hippocampal samples from CMV-nuTRAP mice (*Nrn1* t(10) = 2.486, $p = 0.0322$; *Gad1* t(10) = 0.2409, $p = 0.8145$; *Gfap* t(10) = 0.683, $p = 0.5101$; *AtLAS* t(10) = 9.134, $p = 3.62e-06$). **F** Depletion of genes not expresse in excitatory neurons, and enrichment of genes primarily expressed in excitatory neurons after IP compared to before, in hippocampal samples from CamKIIa-nuTRAP mice (*Gad1* t(5) = 6.65, $p = 0.0012$; *Slc6a1* t(5) = 9.00, $p = 0.0003$; *Gfap* t(5) = 3.69, $p = 0.0141$; *Adora1* t(5) = 2.20, $p = 0.0794$; *Neurod6* t(5) = 3.32, $p = 0.0210$; *Nrn1* t(5) = 1.15, $p = 0.3012$; *Slc17a7* t(5) = 2.87, $p = 0.0350$). **G** Depletion of genes not expressed in inhibitory neurons and enrichment of genes primarily expressed in inhibitory neurons after IP compared to before, in hippocampal samples from vGAT::bacTRAP mice (*Nrn1* t(10) = 10.49, $p = 1.03e-06$; *Slc17a7* t(10) = 11.67, $p = 3.80E-07$; *Gad1* t(10) = 6.23, $p = 9.76E-05$; *Slc6a1* t(10) = 3.01, $p = 0.0131$). **H** Correlation between bulk sequencing and CMV-nuTRAP logFCs 45 mins after stress. Colors indicate significance in analyses (red = significant in both datasets, blue = in CMV-nuTRAP only, yellow = in bulk only). **I** Significant genes that are AS responsive in pooled data from all experiments (time series, CMV-nuTRAP, vGAT::bacTRAP and CaMKIIa-nuTRAP) in the vHC. **J** Significant genes when employing a model that includes interaction terms in the vHC. **K** logFC 45 mins after AS across datasets of the top 20 genes in the vHC. *$p < 0.05$, **$p < 0.01$, ***$p < 0.001$, ****$p < 0.0001$. Male CamKIIa-nuTRAP and vGAT::bacTRAP, and female CMV-nuTRAP were used.

be exclusively expressed in excitatory neurons. (3) Expression of these genes is highly variable in both pyramidal and granular cells along the dorsoventral axis[37,38,65], hence the higher variability lowers the statistical power to show significant enrichment. To confirm the specificity of our TRAP lines we used immunohistochemistry. eGFP was expressed in both neurons and glial cells in CMV-nuTRAP mice (Fig. 6B). Further, we confirmed eGFP and mCherry expression in pyramidal cells in the HC of CaMKIIa-nuTRAP mice (Fig. 6C), and eGFP expression and co-localization with glutamate decarboxylase 65/67 (GAD65/67) in the HC of vGAT::bacTRAP mice (Fig. 6D).

We then subjected CMV-nuTRAP, CaMKIIa-nuTRAP and vGAT::bacTRAP mice to AS, and collected the vHC 45 mins after onset of stress for TRAPseq. We used 4-7 mice per group, and we treated each mouse as an independent biological sample (see Supplementary Fig. 6A for full overview of all experimental designs). TRAPseq analysis revealed that only a small number of genes were significantly changed in either the whole translatome (in vHC), or in the translatome of excitatory or inhibitory neurons in the vHC (Supplementary Fig. 6C) and dHC (Supplementary Fig. 6B). Comparison of the transcriptomic response to that of the CMV-nuTRAP indicates that while large fold-changes on the mRNA level result in corresponding changes in translation, there was no correlation for smaller changes (Fig. 6H). This is in line with "translational buffering" observed in a model of sustained Kainate-induced seizures, where only larger mRNA fold-changes led to translational changes[66]. We indeed observed that the same top regulated genes were increased in translation in both their model and our AS data, albeit at very different magnitudes, in line with the fact that sustained seizures are a much stronger stimulus than our AS exposure (Supplementary Fig. 7E). To ensure that we did not miss translational changes because transcription might not yet have occurred 45 mins after stress, we repeated the Camk2a-nuTRAP experiment 90 mins after stress exposure. Again, the translatomic effects closely mimicked the results obtained at the 45 min time point (Supplementary Fig. 6C), demonstrating the reproducibility of our approach and ruling out that the low number of translated mRNAs is associated with the timing of our analysis. Despite the high number of biological replicates used for our analyses, we noticed that variability in TRAP samples was generally higher than in transcriptomic bulk sequencing analyses, however lower than observed in a previously published study using TRAP in CA3 neurons after stress exposure[67] (Supplementary Fig. 6D). This is in line with a recent reanalysis of several TRAPseq

datasets[68] and suggests that the statistical power of TRAPseq is lower than that of bulk RNAseq. To increase statistical power, we performed a meta-analysis, which included data from both the transcriptome time series assay, as well as all of the TRAPseq samples. We used surrogate variable analysis (SVA)[69] to remove technical variability across batches of samples. Adjusting for baseline differences across datasets and differences in response strength, we identified genes that demonstrated consistent and significant changes across datasets 45 mins after stress (Fig. 6I). These bona-fide stress-induced genes include many well-known immediate early genes such as *Fos, Arc, Sgk1, Egr1, Dusp1* and *Apold1*, indicating that active translation of these genes coincides with increased transcript availability[66]. To detect different stress-induced changes between datasets, we then employed a more complex statistical model with interaction terms (see methods) accounting for these differences. We found numerous genes that were significantly altered (Fig. 6J), most of which also had significant interactions (Supplementary Fig. 6E). This analysis highlights the remarkable cell-type specificity in the translational response to AS. Translation for genes such as *Egr1, Egr2* and *Nr4a1 was* increased in inhibitory neurons, while genes such as *Cyr61* and *Btg2* were translated more in excitatory neurons (Fig. 6K). *Apold1*, an endothelial immediate early gene, was upregulated much less in inhibitory and excitatory neurons compared to the CMV-nuTRAP and bulk RNAseq. Overall, these results demonstrate that different transcriptional and translational programs play out across cell types, and that the initially strong and widespread molecular response is buffered towards protein translation.

**The proteome in different hippocampal subregions after acute stress.** The finding that stress-induced translational changes are much more modest than the phospho-proteomic and transcriptional changes, supports previous findings that hardly any stress-induced proteomic changes can be detected in the dHC and vHC 24 hours after AS[22]. We reasoned that we might observe stronger proteomic changes right after the transcriptional response has played out. While single cell proteomic analyses are not technically possible, proteomic profiles differ dramatically between dHC and vHC[22], and also between CA1 and CA3 subregions[70]. To control for this variability and increase analytical power, we decided to assess proteomic changes 4 h after the initiation of stress separately in the CA1, CA3 and DG areas of both the dHC and vHC (i.e., six regions per mouse), using data-independent label-free proteomics (Fig. 7A).

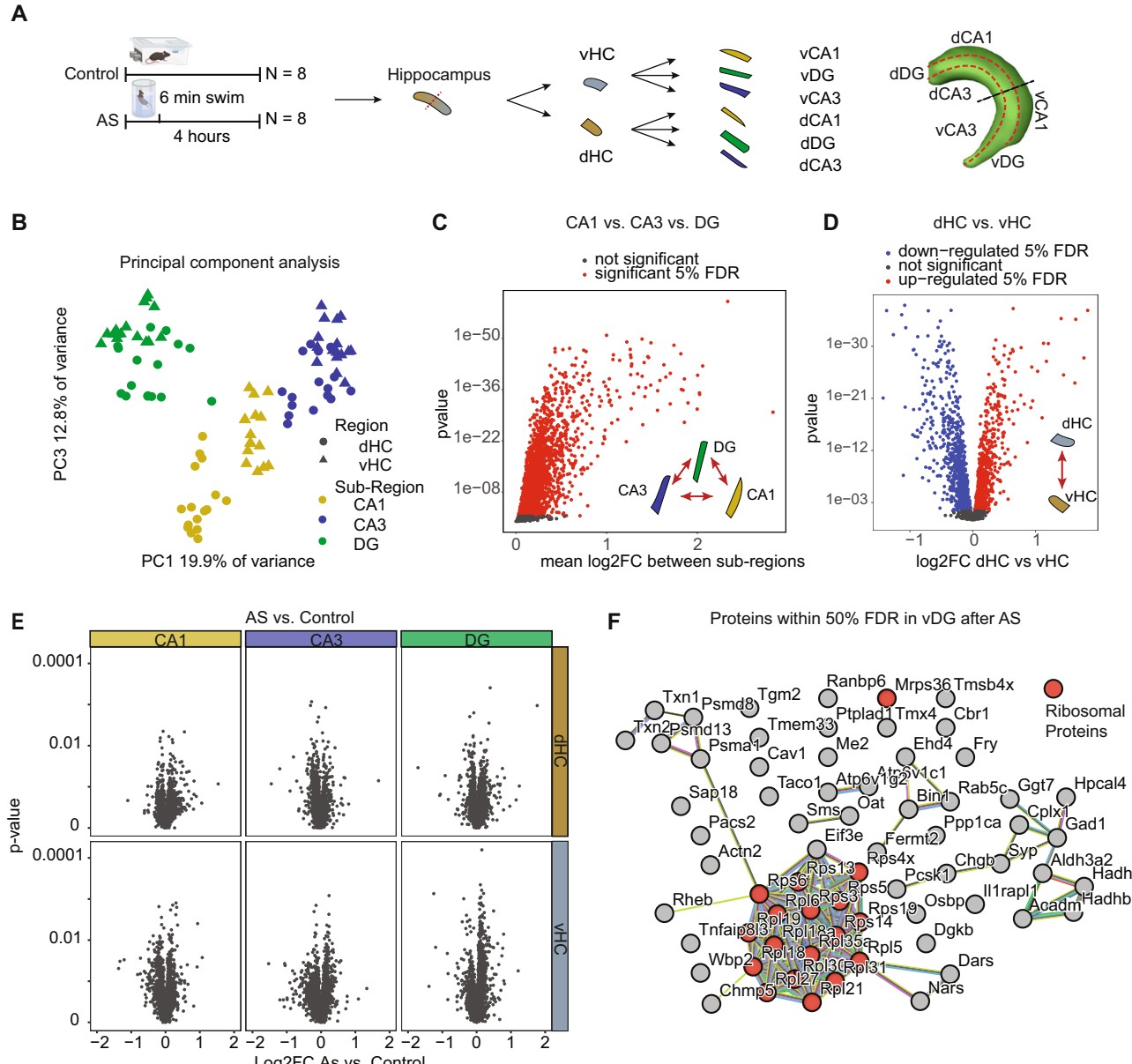

**Fig. 7 The effects of acute stress (AS) on the proteome of CA1, CA3 and DG in the dorsal hippocampus (dHC) and ventral hippocampus (vHC).**
**A** Experimental design and tissue collection approach. **B** Principal component (PC) analysis of all samples. The first component resolves sub-regional differences, whereas the 3rd further resolves dorsal-ventral differences. We did not find any component that resolves stress. **C** Statistical results for ANOVA subregion * region within animals (omitting stress vs homecage as variable). Many proteins have significant (red, within 5% false discovery rate (FDR)) sub-regional expression differences at baseline. **D** Same analysis as in (**C**). Many proteins show significant dorsoventral expression differences at baseline (red, upregulated in dHC 5% FDR, blue downregulated in dHC 5% FDR). **E** Volcano plots showing the log2 fold change and statistical results of the 2 group comparisons homecage vs swim of proteins 4 h after a 6 min exposure to AS in all regions and sub-regions. No proteins are within the 5% FDR. **F** protein-protein interaction network of stress sensitive proteins in the vDG (50% FDR). Experiments performed in male mice.

A principal component analysis showed that both sub-regional differences as well as dorsal-ventral differences could be resolved on the proteomic level, confirming strong baseline differences between proteomic profiles of all regions (Fig. 7B). Statistical analyses revealed that most proteins (N = 2536 of 3039) had profound sub-regional expression differences between CA1, CA3 and DG (Fig. 7C). Further, we found that roughly half of all proteins (N = 1762) had different baseline expression levels between the dHC and vHC (Fig. 7D). Overall, this demonstrates that strong sub-regional and dorsal-ventral baseline differences in protein expression are present, which confirms and extends our previous proteomic work[22,70], and is in line with the well-

characterized transcriptional heterogeneity across hippocampal subregions[37,38,71,72]. Despite these clear differences between subregions, we did not detect any significant stress-induced changes in protein expression in any of the subregions (Fig. 7E, FDR < 0.05). To compensate for a slightly higher biological coefficient of variation observed with proteomics compared to transcriptomics (Supplementary Fig. 7A–C), we used larger group sizes for the proteomic analyses. This resulted in similar power for both methods (Supplementary Fig. 7D), suggesting that the absence of stress-induced proteomic changes is not a result of low statistical power. To perform an exploratory analysis, we relaxed the significance cutoff to a FDR of 0.5, which revealed subtle

changes exclusively in the DG of the vHC, and a protein-protein interaction analysis showed an enrichment for ribosomal proteins (Fig. 7F).

## Discussion

Here we present an in-depth, multiomic characterization of the acute stress (AS) response in the mouse HC. Our results reveal a pronounced molecular response, which starts with an immediate and broad wave of protein phosphorylation that resolves rapidly between 15 to 30 mins after the onset of stress. This is followed by strong active nuclear transcription that quickly resolves between 45 to 90 mins after stress. Waves of gene expression peak between 45 to 90 mins and completely resolve within 3 to 4 hours. A substantial reduction of mRNAs also occurs during this period, which appears to be largely due to mRNA decay. Using snRNAseq, we show that many genes have remarkable cell-type specificity, and at 45 minutes after stress glial cells contribute most to the transcriptional bulk sequencing signal. Surprisingly, only a subset of differentially expressed genes become actively translated, and again the translational response varies markedly between different cell types. At the level of the proteome, we detect profound baseline differences between subregions of the HC, but hardly any stress-induced changes. From phosphorylation to translation, the response to stress is very similar between males and females. Although the phosphorylation changes are more pronounced in females than in males, these differences seem to decrease as the stress response reaches transcription and translation, in line with previous transcriptional and translational analyses[23,68]. The observation that the molecular stress response is tightly controlled and terminates efficiently, is in line with the short-lived increase in anxiety levels we observe on a behavioral level. However, a memory trace of the stressful experience and more subtle behavioral changes likely remain, and more targeted analyses at the level of synapses or cellular engrams might reveal longer-lasting molecular changes.

Since stress can differentially impact the dHC and vHC[22,73,74], we conducted most of our analyses in both fractions of the HC. Although we find extensive baseline differences between the dHC and vHC across all molecular levels, we observe a rather homogeneous stress-response in both regions. Overall, molecular changes can differ in magnitude between these regions, however changes are usually strongly correlated, suggesting coordinated molecular responses that unfold throughout the HC. This is in line with a recent meta-analysis, which revealed that the molecular response to AS is rather uniform across the dorsal-ventral axis[2]. Although our data were exclusively obtained after forced swim stress, the transcriptional response we observe in both dHC and vHC is remarkably similar to the changes previously described after novelty stress or restraint stress[2,22]. Thus, although some of our effects are likely modality-specific, we expect that a significant proportion of our results will also translate to different acute stress models.

To our knowledge, this is the first description of phospho-proteomic changes after AS. Our employed methods do not allow separate analyses of different cell types, which remains a major challenge in neuroproteomics[75]. However, overrepresentation analysis suggests that a large part of the response occurs in neurons. In agreement with previous literature, we find widespread stress-induced changes in pathways related to dendritic morphology, synaptic function and plasticity, like the calcium-dependent neuronal MAPK/ERK pathway[30,76,77]. We observed increased phosphorylation of MAPK2, P38 MAPK, CAMKII, and SYN1, which are linked to synaptic plasticity[30,78,79]. Notably, SYN1 can stay phosphorylated up to 24 hours after an intense, inescapable 40 min footshock stressor[35]. The fact that we did not detect any lasting effects beyond 30 mins could indicate that the response to brief AS can be effectively terminated, but that more intense stressors may dysregulate the

system, a hypothesis that needs to be investigated. Beyond known pathways, we identify large numbers of phosphorylated proteins that - to our knowledge - have never been reported to change after an AS exposure. Notable candidates include the proteins SHISA6, SHISA7, SHANK1, SHANK2 and SHANK3, which are related to postsynaptic density and plasticity[80,81].

Although previous work has assessed the transcriptomic changes after AS (for review see[2]), our results describe the temporal dynamics of the transcriptional response from start to end. We find groups of genes with specific temporal response profiles. Most gene clusters that peak early (45 mins) after stress are involved in the regulation of transcription and intracellular signaling cascades (e.g., regulation of MAPK activity). This is in line with earlier reports that the MAPK/ERK pathway regulates transcription by inducing *Fos* and *Egr1* in the HC[82–84]. In contrast, groups of genes that are regulated at 90 mins are not only involved in transcriptional regulation but also in metabolic processes, such as maintaining the internal state of glucose. This makes sense, considering that the key purpose of the stress response is to supply the energy resources necessary to meet heightened metabolic demand[2,85]. Previous in vitro data had shown that application of the stress hormone corticosterone can trigger gene expression changes that last up to 5 h. Our data show that all gene expression changes dissipate between 3–4 hours after stress, suggesting that regulation of transcription is very tightly controlled in vivo. This agrees with the notion that an efficient termination of the stress response is crucial for recovery[2,4,86].

It is known that mRNA levels do not always correlate well with the levels of the corresponding proteins[22,87,88]. In fact, evidence suggests that the control of protein abundance predominantly happens at the level of translation[52]. To address this, we employed TRAPseq, which provides cell type specificity, avoids cell dissociation, minimizes the loss of RNAs that are localized in dendrites or axons, and allows a high depth of sequencing[88,89]. We extend previous work that has used TRAPseq to reveal the translatomic response to AS specifically in excitatory neurons of the CA3 region[67], by providing a global translatome profiling after stress, and a comparison of the translational profile for inhibitory and excitatory neurons of the dHC and vHC. In agreement with previous reports[83,90], *Egr1* and *Fos* were among the top upregulated targets, however *Egr1* was upregulated much more strongly in inhibitory neurons. Similarly, we found *Egr2* to be induced by stress mainly in inhibitory neurons, but in contrast *Cyr61* is strongly upregulated in excitatory neurons. In general, however, we detect far fewer stress-induced changes in the translatome than expected based on our transcriptomic data. Importantly, our bona-fide stress-induced genes (Fig. 6I) strongly agree with our recent meta-analysis of TRAPseq data[68], which gives us confidence that our analysis accurately captures the translatomic stress response.

Data integration across molecular levels poses a number of challenges, beginning with differences in detection rates and sensitivity of the technologies. A striking example is the disconnect between transcriptomics and proteomics. Of all 1,264 genes differentially altered on the transcriptional level, we were only able to quantify 105 corresponding proteins (lower than the expected overlap of 294 proteins). This is in part due to the fact that many of the significantly altered TFs are notoriously hard to detect using proteomics, owing to their low abundance and nuclear localization. In addition, the datasets reflect different temporal dynamics. For example, our snRNAseq data provide a snapshot of active transcription at 45 mins after stress, which is well recapitulated by the unprocessed bulk transcripts at the same time point, whereas the transcriptome (processed mRNAs) chiefly reflects the accumulation of previous transcriptional waves, and the translatome presumably reflects longer-lived mRNAs that become translated.

Despite these challenges, some themes can be observed across different datasets. First, the earliest changes in phosphorylation

are very strong, decreasing in intensity as the cascade moves from transcription to translation. This is reminiscent of translational repression as it occurs in the context of the cellular "integrated stress response"[91,92]. How the psychophysical stress response we describe here compares to the "integrated stress response" needs to be carefully addressed in future work. Second, multiple indications suggest that part of the molecular cascade is driven by MAPK/ERK and CREB (Creb1) activity in neurons. Starting at the level of protein phosphorylation, we noticed enrichment for many neuronal proteins, as well as calcium signaling and MAPK/ERK activity. Other studies have described activity-dependent calcium signaling in neurons in great detail[93,94] and demonstrated a clear link to the transcription factor Creb1[95,96]. While we were not able to detect CREB1 directly in our phosphorylation data, we detected an enrichment for Crebbp activity in the early transcriptome, a protein that is directly linked to Creb1-mediated transcription factor activity. Further, our transcriptome data fully recapitulate the first wave of rapid primary response genes that were shown to depend on ERK/MAPK signaling in cell culture experiments[50]. Additionally, in both inhibitory and excitatory neurons our TRAP data identify active translation in some candidate genes that are targets of CREB1, such as *Egr1*, *Btg2*, *Junb* and *Dusp1*[97–99]. These data suggest that stress initially triggers calcium-induced changes in synapses that rapidly regulate neuronal gene expression via ERK/MAPK and CREB pathways. Interestingly, subsequent gene expression changes are poorly explained by neuronal activity. This is shown by the fact that the early transcriptional response to AS fully recapitulates transcriptional changes observed after neuronal activation in vitro, while the late response strongly differs between AS in vivo and neuronal activation in vitro (Supplementary Fig. 4E). In line with this, our snRNA data also indicate that a large portion of the active transcriptional response (measured 45 mins after stress) is driven by non-neuronal cell types, particularly vascular cells and astrocytes. Indeed, a recent translatome screen in astrocytes from the somatosensory cortex revealed robust translational changes 90 mins following acute restraint stress[100]. Thus, the different waves of transcriptional activity could to some degree be dependent on different cell populations and other pathways that are impossible to stratify in bulk sequencing data. To further explore how the dynamics of the AS play out over time across different cell types, single-cell data with a higher temporal resolution will be required.

**Summary and Outlook**. Our work provides a detailed characterization of the molecular cascades that unfold as part of the healthy acute stress response. It also lays the foundation to further explore how stressors of different severity or duration impact different cell types and the metabolic profile of brain structures involved in mediating the stress response. To facilitate this process, all our data are publicly accessible and searchable through an interactive app. Future studies will need to test whether more severely stressful experiences, or chronic stress exposures, lead to more exaggerated, less tightly regulated transcriptional and translational consequences, and whether such dysregulation would be associated with stress-related psychiatric disorders.

## Methods

**Animals**. All experiments were conducted in accordance with the Swiss federal guidelines for the use of animals in research and under license ZH161/17, approved by the Zurich Cantonal veterinary office. For experiments with wild type animals, C57Bl/6J mice were either obtained from Janvier (France) or bred at the ETH Zurich animal facility (EPIC). For the generation of CMV-nuTRAP mice, heterozygous CMV-Cre mice (B6.C-Tg(CMV-cre)1Cgn/J, MGI:3613618) were bred with homozygous or heterozygous floxed nuTRAP mice (B6;129S6-Gt(ROSA) 26Sor$^{tm2(CAG-NuTRAP)Evdr}$/J, MGI:104735). For the generation of CaMKIIa-nuTRAP mice, homozygous floxed nuTRAP mice (B6;129S6-Gt(ROSA)

26Sor$^{tm2(CAG-NuTRAP)Evdr}$/J, MGI:104735) were bred with homozygous CaMKIIA-Cre (B6.Cg-Tg(Camk2a-cre)T29-1Stl/J, MGI:3613616) mice. Mice were housed in groups of 4-5 per cage in a temperature- and humidity-controlled facility on a 12-hour reversed light-dark cycle (lights off: 9:15 am; lights on: 9:15 pm), with food and water ad libitum, and used for experiments at the age of 2-5 months. All experiments were conducted during the animals' active (dark) phase. Experiments were conducted with either male or female mice. For all experiments, mice were single-housed 24 hours before exposure to stress, which we have previously shown to reduce corticosterone levels in both sexes, and which also avoids confounding gene expression effects that result from disturbing cage mates on the test day[23,101].

**Open field test (OFT)**. Open-field testing was performed in sound insulated and ventilated multi-conditioning chambers (MultiConditioning System, TSE Systems Ltd, Germany), as described previously[29,102]. Briefly, the open field had dimensions 45 cm (l) × 45 cm (w) × 40 cm (h), and consisted of four transparent Plexiglas walls and a light gray PVC floor. Mice were tested under dim lighting (4 Lux) 75 dB of white noise was playing through the speakers of each box. Mice were placed directly into the center of the open field and the tracking/recording was initiated upon the first locomotion grid beam break. The test lasted 10 mins.

**Elevated Plus Maze (EPM)**. The EPM was made from gray PVC, with arms measuring 65.5 × 5.5 cm ($L × W$), elevated 61.5 cm. Light intensity in the open arms was at 19–21 lux. All EPM tests were 5 mins in duration. Distance, velocity, time in zone (open/closed arms + center) and head dips were Using the "EPMAnalysis" function of the "DLCAnalyzer" package[102].

**Swim stress paradigm and tissue collection**. For cold swim stress, mice were placed for 6 mins in a plastic beaker (20 cm diameter, 25 cm deep) filled with 18 ± 0.1 °C water to 17 cm, in a room with dim red lighting. Immediately after stress exposure, mice were returned to their assigned single-housing homecage. At the appropriate time point after initiation of stress (for immediate groups within maximum one min after offset of stress), mice were euthanized by cervical dislocation and decapitation. The brain was quickly dissected on PBS-ice and isolated hippocampi were either snap-frozen in liquid nitrogen and stored at −80 °C until further processing, or transferred to a binocular microscope and cooled with PBS-ice for dissection of subregions (proteomics). Area CA3 was dissected by cutting along the minor hippocampal fissure along the dorsoventral axis of the hippocampus. Area CA1 and DG were separated with a pincer and incision scalpel by gently pushing the blade along the major hippocampal fissure prior to snap-freezing and storing at −80 °C[70].

## Proteomics

*Protein extraction*. We used a block design for sample processing. Samples were split into multiple blocks, containing one replicate of each condition and subregion/region. Processing order within blocks was randomized, as was the order of blocks. Proteins were extracted from pooled hemispheres of dHC or vHC CA1, CA3 and DG samples using 150 µL TEAB buffer (100 mM triethylammonium bicarbonate, 0.1% SDS, 1:100 protease inhibitor cocktail P8340 (Sigma-Aldrich, St. Louis MO, USA), 1:500 PMSF (50 mM in EtOH)). Samples for phosphoproteomics (pooled hemispheres of dHC/vHC) were extracted using the same buffer with the addition of 1:100 phosphatase inhibitor cocktail 2 (P5726, Sigma-Aldrich, St. Louis MO, USA), and phosphatase inhibitor cocktail 3 (P0044, Sigma-Aldrich, St. Louis MO, USA).

The samples were mechanically lysed by 15 strokes with a 26 G needle and sonicated for 5 mins. Samples were spun down at 16,000 g for 30 mins (1 hour for phospho samples) at 4 °C and supernatants were collected. Proteins were quantified using a Qubit protein assay kit (ThermoFisher Scientific, Waltham MA, USA) following the manufacturer's protocol. Protein extracts were further processed with a filter assisted sample preparation protocol[103]. 20 µg of protein were added to 30 µL SDS denaturation buffer (4% SDS (w/v), 100 mM Tris/HCL pH 8.2, 0.1 M DTT). For phospho samples 300 µg of protein were added to equal amount (v/v) of concentrated denaturation buffer (0.2 M DTT). For denaturation, samples were incubated at 95 °C for 5 mins. Samples were diluted with 200 µL UA buffer (8M urea, 100 mM Tris/HCl pH 8.2) and then loaded to regenerated cellulose centrifugal filter units (Microcon 30, Merck Millipore, Billercia MA, USA). Samples were spun at 14000 g at 35 °C for 20 mins. Filter units were washed once with 200 µL of UA buffer followed by centrifugation at 14,000 g at 35 °C for 15 mins. Cysteines were alkylated with 100 µL freshly prepared IAA solution (0.05 M iodoacetamide in UA buffer) for 1 min at room temperature (RT) in a thermomixer at 600 rpm followed by centrifugation at 14000 g at 35 °C for 10 mins. Filter units were washed three times with 100 µL of UA buffer then twice with a 0.5 M NaCl solution in water (each washing was followed by centrifugation at 35 °C and 14000 g for 10 mins). Proteins were digested overnight at RT with a 1:50 ratio of sequencing grade modified trypsin (0.4 µg for whole proteome samples, 6 µg for phospho samples, V511A, Promega, Fitchburg WI) in 130 µL TEAB (0.05M Triethylammoniumbicarbonate in water). After protein digestion overnight at RT, peptide solutions were spun down at 14000 g at 35 °C for 15 mins and acidified with 10 µL of 5% TFA (trifluoroacetic acid). 100% ACN was added for a final concentration of 3% in the samples. Phospho-samples were split into reference

samples (1:20 of final amount, used for peptidic clean-up) and enrichment samples (19:20 of final amount, used for phospho enrichment).

*Enrichment of phospho peptides.* For phospho enrichment digested samples were first lyophilized and resuspended in 200 µL loading buffer (1 M glycolic acid in 80% ACN, 5% TFA). Phosphopeptides were enriched using MagReSyn® Ti-IMAC magnetic beads (Resyn Biosciences). 20 µL of beads were used per sample. Microspheres were washed once with 400 µL of 70% EtOH for 5 mins at RT and reactivated using 200 µL of resuspension solution (1% NH4OH) for 10 min. Microspheres were then equilibrated twice with 200 µL loading buffer for 1 min and then resuspended in 500ul 100% ACN per 20ul of beads. A Thermo Scientific™ KingFisher™ was used for enrichment of phospho peptides. First, microbeads were loaded onto magnetic rods and washed with 500 µL loading buffer. Samples were loaded and washed in 500 µL loading buffer, in 500 µL wash 2 buffer (80% ACN, 1% TFA) and in 500 µL wash 3 buffer (10% ACN, 0.2% TFA). Samples were then eluted in 200 µL resuspension solution (1% NH4OH) and lyophilized in a speedvac then re-solubilized in 19 µL 3% ACN/0.1% FA (formic acid) prior to LC-MS/MS measurements. 1 µL of synthetic peptides (Biognosys AG, Switzerland) were added to each sample for retention time calibration.

*Peptide clean-up.* Digests of non-phospho samples were cleaned-up using StageTip C18 silica columns (SP301, Thermo Scientific, Waltham MA). Columns were conditioned with 150 µL methanol followed by 150 µL of 60% ACN (acetonitrile)/ 0.1% TFA. Columns were equilibrated with 2 x 150 µL of 3% ACN/0,1% TFA. Samples were loaded onto the columns. They were then washed with 2 x 150 µL 3% ACN/0.1% TFA and eluted with 150 µL 60% ACN/0.1% TFA. Samples were lyophilized in a speedvac then re-solubilized in 19 µL 3% ACN/0.1% FA (formic acid) prior to LC-MS/MS measurement. 1 µL of synthetic peptides (Biognosys AG, Switzerland) were added to each sample for retention time calibration.

*LC-MS/MS measurements.* Samples were measured on a Q Exactive HF (Thermo Fisher Scientific, Waltham MA, USA, using Xcalibur v4.3). Peptides were separated with an ACQUITY UPLC M-Class System (Waters, Milford MA, USA). We used a single-pump trapping 75-µm scale configuration (Waters, Milford MA, USA). 1 µL of each sample (4 µL for phospho samples) were injected. Trapping was performed on a nanoEase™ symmetry C18 column (pore size 100 Å, particle size 5 µm, inner diameter 180 µm, length 20 mm). For separation a nanoEase™ HSS C18 T3 column was used (pore size 100Å, particle size 1.8 µm, inner diameter 75 µm, length 250 mm, heated to 50 °C). Peptides were separated using a 120 min long linear solvent gradient of 5–35% ACN, 0.1% FA or a 60 min long linear gradient for phospho peptides (both using a flowrate of 300 nL/min). Electronspray ionization with 2.6 kV was used and a DIA method with a MS1 in each cycle followed by 35 fixed 20 Da precursor isolation windows within a precursor range of 400-1100 m/z was applied. For MS1 we used a maximum injection time of 55 ms and an AGC target of 3e6 with a resolution of 30 K in the range of 350–1500 m/z). MS2 spectra were acquired using a maximum injection time of 55 ms an AGC target of 1e6 with a 30 K resolution in the range of 140 − (2 x the upper range of the precursor window) m/z. A HCD collision energy of 28 was used for fragmentation.

*Peak picking and quantification of non-phospho samples.* We used Spectronaut™ (Biognosys, version 10) with directDIA for peak picking and sequence assignment. We used a *Mus musculus* reference proteome for C57BL/6J from uniprot (UP000000589) from the Ensembl GCA_000001635.8 assembly only including reviewed entries. We included a maximum of 2 missed cleavages, using a Tryptic specificity (KR/P). Sequences in a range of 7-52 AA were considered. We included carbamidomethyl as fixed modification for cysteine, oxidation as variable modification for methionine and protein N-terminal acetylation as variable modification. A maximum of 5 variable modifications were considered. Single hit was determined on the stripped sequence level. Major grouping was done by protein group ID and minor grouping by stripped sequence. Only proteotypic peptide sequences were considered and single hit proteins excluded. For the minor and major group quantification the top 3 entries were used using the mean precursor/peptide quantity. A localized normalization strategy and interference correction were used. Machine learning was performed on a per run basis and iRT profiling was enabled.

*Peak picking and quantification of phospho samples.* For phospho-samples we used the following additions/modifications to the Spectronaut™ searches. We additionally included phosphorylation of serine, threonine or tyrosine as variable modification, either as full group or following a neutral loss of 98. PTM localization was enabled with a probability cutoff of 0.75 and quantification performed on the precursor level. We included both proteotypic and non-proteotypic peptides. A global median normalization strategy was used.

*Statistical analysis of LC-MS/MS data.* For non-phospho samples we used exported quantity values of proteins from Spectronaut. For statistical analysis we used R version 3.6.3. We performed two group analyses (Homecage vs Swim) within subregions and regions. We used log2 transformed quantity values and performed an ANOVA analysis (using the aov() of function R) whilst correcting for block effects. PCA was performed on the expression matrix of all samples. For subregion

vs. region analysis we used a two-way ANOVA with the formula sub-region * region while correcting for overall effects within animals.

For phospho measurements quantity values were exported from Spectronaut on the modified peptide level. We then used the R package "DEP" (version 1.6.1) for differential analysis of defined contrasts. For the missing value filter we used a threshold of 2 and for the imputation we used the "MinProb" method with a $q = 0.01$.

For the male vs. female phosphoproteomic analysis Spectronaut™ version 11 was used. Data imputation was done directly with spectronaut and for statistical analysis aggregated precursor intensities of modified peptides were log2 transformed prior to t-test (2 group comparisons) and two-way ANOVAs (sex * swim analyses). For multiple testing correction the Benjamini–Hochberg false discovery rate (FDR) method was used.

### Transcriptomics and translatomics

*Whole tissue RNA extraction.* We used a block design for sample processing. Samples were split into multiple blocks, containing one replicate of each condition and region and processing order within blocks randomized. Samples were homogenized in 500 µL Trizol (Invitrogen 15596026) in a tissue lyser bead mill (Qiagen, Germany) at 4 °C for 2 mins, and RNA was extracted according to manufacturer's recommendations. RNA purity and quantity were determined with a UV/V spectrophotometer (Nanodrop 1000), while RNA integrity was assessed with high sensitivity RNA screen tape on an Agilent Tape Station/Bioanalyzer, according to the manufacturer's protocol. The RIN values of all samples ranged from 8.4 to 10.0. For library preparation, the TruSeq stranded RNA kit (Illumina Inc.) was used according to the manufacturer's protocol.

*Processing of TRAP samples.* To isolate actively translated RNA from specific cell types in the HC, the TRAP protocol was followed as described in[60]. The samples were homogenized with a 2 mL tissue grinder and a pestle (3432S90, Thomas Scientific), on a Rotor for homogenizers (LT-400D, Yamato) in tissue lysis buffer. Solutions and the affinity matrix were prepared as recommended, and optimal bead titers were determined for every transgenic line with affinity matrix titration. More specifically, for samples of both the dHC and vHC of CaMKIIa-nuTRAP mice, the affinity matrix was prepared using 75 µL of Streptavidin MyOne T1 Dynabeads (65601, Invitrogen) per sample. For the vGAT::bacTRAP mice 75 µL beads per sample were used for the dHC and 150 µL for the vHC, and for the vHC samples of the CMV-nuTRAP mice 300 µL beads were used per sample. The corresponding volumes of biotinylated protein L and the GFP antibodies 19C8 and 19F7 were added, and the affinity matrix (in a final volume of 200 µL) was added to every sample for immunopurification. Samples were incubated at a tube rotator at 4 °C for roughly 17 hours.

After immunopurification we proceeded to RNA elution and purification with the RNeasy Plus Micro Kit (74034, Qiagen). The RNA was first dissociated from the ribosomes/beads with 350 µl Buffer RLT Plus from the kit, containing 40 µM dithiothreitol (DTT, R0861, Thermo Fisher Scientific), and purified according to the manufacturer's instructions. RNA concentration was estimated with either the RiboGreen fluorescence assay (R11491, Thermo Fisher Scientific) and a NanoDrop 3300 microvolume fluorospectrometer, or with the Quantifluor RNA System (E3310, Promega) and a DS-11 Series Spectrophotometer / Fluorometer (DeNovix). The integrity of the RNA was assessed using the RNA 6000 Pico Kit (5067-1513, Agilent Technologies) on a 2100 Bioanalyzer (Agilent Technologies Inc., CA), and all RIN values were between 8.2 and 10.0. Conversion to cDNA and generation of libraries were performed with the Smart-seq2 protocol[61].

*Bulk and TRAP sequencing and data analysis.* Library preparation and sequencing was performed at the Functional Genomics Center Zurich (FGCZ) core facility. For bulk sequencing library preparation, the TruSeq stranded RNA kit (Illumina Inc.) was used according to the manufacturer's protocol. The mRNA was purified by polyA selection, chemically fragmented and transcribed into cDNA before adapter ligation. For bulk and TRAP sequencing single-end sequencing (100nt) was performed with Illumina Novaseq or HiSeq 4000. Samples within experiments were each run on one or multiple lanes and demultiplexed. A sequencing depth of ~20M reads per sample was used. Adapters were trimmed using cutadapt[104] with a maximum error rate of 0.05 and a minimum length of 15. Kallisto[105] was used for pseudo alignment of reads on the transcriptome level using the genecode.vM17 assembly with 30 bootstrap samples and an estimated fragment length of 200 ± 20. For differential gene expression (DGE) analysis we aggregated reads of protein coding transcripts and used R (v. 3.6.2) with the package "edgeR" (v 3.26.8) for analysis. A filter was used to remove genes with low expression prior to DGE analysis. EdgeR was then used to calculate the normalization factors (TMM method) and estimate the dispersion (by weighted likelihood empirical Bayes). For two group comparisons the genewise exact test was used, for more complex designs we used a generalized linear model (GLM) with empirical Bayes quasi-likelihood *F*-tests. For multiple testing correction the Benjamini–Hochberg false discovery rate (FDR) method was used.

To quantify spliced (i.e. exonic) and unspliced transcripts, we aligned the reads using STAR[106] and ran featureCounts[107] with three sets of parameters: i) using exons as features (standard), ii) using exons with --nonSplitOnly, and iii) using transcripts as features with --nonSplitOnly (all three additionally shared the "-O

--largestOverlap --nonOverlap 3 --fracOverlap 0.9 --primary -s 2" parameters). The first was used as a quantification of processed transcripts; for unprocessed transcripts, we subtracted from the third the counts of the second (which are compatible with processed transcripts as well).

For analyses of datasets originating from multiple experiments (i.e. different TRAP protocols) we further employed SVA correction to correct for processing specific effects[69]. Surrogate variables independent of experimental groups were identified using the sva package 3.34.0 on data after DESeq2 variance-stabilization[108], and were then included as additive terms in the GLMs.

For the metaanalysis of TRAP data across datasets (Fig. 6I/J) we used the following SVA correction models and subsequent GLM models. Additive analysis; SVA model: ~Dataset + Condition, GLM model ~SVs + Condition. Interactive analysis; SVA model: ~Dataset * Condition, GLM model: ~SVs + Condition * Dataset. SVs = surrogate variables.

Heatmaps were produced with the SEtools package. To avoid rare extreme values from driving the scale, the color scale is linear for values within a 98% interval, and ordinal for values outside it. Unless otherwise specified, the rows were sorted using the features' angle on a two-dimensional projection of the plotted values, as implemented in SEtools.

*Single nucleus RNA sequencing and analysis.* Tissue dissociation: All steps for tissue dissociation were performed on ice. Both hippocampi of the same animal were chopped briefly in 100 μL lysis buffer (10 mM Tris-HCl, 10 mM NaCl, 3 mM MgCl2, 0.1% Igepal, 0.2 U/ul recombinant RNase inhibitor in nuclease-free water) and immediately transferred to a dounce homogenizer filled with 1 mL lysis buffer. Tissue was homogenized by 15–20 strokes with each pestle A and pestle B. 2 mL of lysis buffer was added to the dounce homogenizer and the solution was incubated for 5 mins. The dounce homogenizer was filled with 4 mL wash buffer (1% BSA. 0.2 U/ul recombinant RNase inhibitor in PBS) and homogenate was passed through a 40 μm filter before adding 8 mL wash buffer. The sample was centrifuged at 500 g for 5 mins at 4 °C and the pellet was resuspended in 7 mL nuclei suspension buffer (1% BSA, 0.2 U/μl RNaseInhibitor (Superasin) in PBS). Then the sample was centrifuged at 500 g for 5 mins at 4 °C. The pellet was resuspended in 1 mL nuclei suspension buffer and filtered through a 40 μm, 30 μm and 10 μm filter. Nuclei count was assessed manually by Trypan blue staining and adjusted to a final concentration of 1000 nuclei/μL.

Library preparation and sequencing: cDNA synthesis and library preparation were conducted following the Chromium Single Cell 3' Reagent Kit v3 User Guide (CG000183 Rev B, 10X Genomics). 14 PCR cycles were performed for cDNA amplification. Each library was sequenced on a NovaSeq6000 instrument aiming for a sequencing depth of 40'000 reads/nucleus.

For data analysis, reads were demultiplexed, aligned and quantified with Cell Ranger 4.0.0[107] using the GRCm38/GENCODE M23 pre-mRNA transcriptome from 10x. The code for downstream analysis is available in the repository. In brief, we first decontaminated the cells from ambient RNA using *SoupX* 1.4.5[108], using cell clusters obtained from the "fastcluster" function of *scDblFinder 1.4.0*. We then removed doublet identified by *scDblFinder*[109], and additionally removed cells (using *scater 1.18.0*[110]) which had more than 9% mitochondrial reads or departed from the median by more than 3 median absolute deviations on either of the following cell QC metrics: log(UMI counts), log(features detected), or percent of UMI in the top 50 features (filtering out higher only). We next normalized the data using *sctransform 0.3.1*[111], and ran PCA and clustering at various resolutions using *Seurat 3.2.2*[112]. For each resolution, we then trained an *xgboost* classifier (10-fold cross-validation at 7:3 train:test split) using the top 1000 genes to estimate the clusters' stability and identifiability, identifying a Leiden resolution parameter of 0.25 as providing an appropriate resolution with low misclassification rate.

The clusters showed very distinct profiles for known markers of broad cell classes. To refine the annotation, we applied cell-level assignment using *scClassify 1.2.0*[113] (ensemble approach, using limma, BI and DD for feature selection, and Pearson, Spearman and Cosine as similarity metrics) based on the HCL1 reference from[114]. There was overall a good agreement between our clusters and the cell-level assignments (in particular separating dentate gyrus neurons from other excitatory neurons), except for subtypes of vascular cells (which were represented in small numbers in our data). For this reason, for downstream analysis we kept this resolution for these cells. One cluster had cells with very low library sizes and showed expression of both neuronal and astrocytic markers, and was consequently excluded; in addition, two clusters with very low abundance had conflicting marker expression and were not represented across all samples; they were also excluded.

As preliminary differential expression analyses did not show significant differences in the response of the two astrocyte clusters or of the three excitatory neuron clusters, they were respectively merged as Astrocytes and Excitatory Neurons. Differential expression analysis was then performed using *muscat 1.4.0*[115] pseudo-bulk method. To assign bulk DEGs to a cell type on the basis of the single-nuclei data, we first calculated the absolute (pseudo-bulk) difference between stressed and control animals for each cell type, weighted it by multiplying it by the power of 1 minus the *p* value in the cell type, and identified the cell type with the highest weighted difference in the same direction as the bulk change. When the absolute weighted difference in the top cell type was less than twice as high as the next cell type, the gene assigned was considered ambiguous.

*Transcription factor activity inference.* Transcription factor regulons were obtained from[57], lifted to the mouse genome using ensembl orthologs, and the likelihood of the interactions was weighted according to the approximative AUC of the different categories in their benchmark. Category E interactions were discarded, and transcription factors that did not pass differential expression filtering were excluded. We next calculated sample-wise activity score per transcription factor in the unspliced data using *viper* 1.24 with pleiotropy correction[116]. To identify factors with differences in activity, we then fitted ~region+time point models on each factor and tested for dropping the time point coefficients using *limma* 3.46[117]. Only factors with an FDR < 0.005, an absolute log-fold change >0.2 and a mean log(counts per million) >= 2 were considered.

To co-cluster differentially-expressed genes based on both regulons and expression, we first produced a TF-target matrix multiplying the likelihood of the interaction with the factor's mode of action, and performing z-score normalization on each factor, and merging this matrix with the genes' log-fold changes at 45 mins, 90 mins and 2 hours, in both unspliced and spliced fractions. Partitioning around 9 medoids was used to cluster genes based on this merged information. For each cluster, gene ontology enrichment analysis was performed with *goseq* 1.41[118] using the tested genes as background, and the top terms were reported after eliminating entirely redundant terms. Overlayed protein-protein interactions between the selected factors were obtained from the BioGrid 4.2.193[119].

*Small RNA sequencing and data analysis.* 1000 ng of total RNA was used as input for Nextflex v3 small RNA library preparation (Perkin Elmer Applied Genomics). Adapters and primers were undiluted and libraries were amplified with 15 PCR cycles prior bead-based size selection. Libraries were prepared in 3 batches composed of samples of both hippocampal subregions and stressed/non-stressed groups. All resulting libraries were pooled to be equimolar and sequenced on 2 lanes of an Illumina Genome Analyzer HiSeq 2500 (Illumina) with a single ended read length of small 50 bp. The experimenter was blinded to the treatment.

For data analysis, reads were first trimmed using cutadapt (adapter sequence TGGAATTCTCGGGTGCCAAGG, 0.05 error rate, minimum length of 18). Reads with the same sequence (including UMI) were then collapsed, and the UMIs trimmed. Quantification was then performed using Sports 1.0[120], using the software's full mouse annotation as described. SVA analysis was used to identify two vectors of technical variation which were then included, along with the hippocampal region, as covariates in the differential expression model. Differential expressions testing was done with edgeR using QLFit, testing for the dropping of the condition and region:condition coefficients.

**K-means clustering**. Analysis of expression profiles was performed using k-means clustering. A matrix listing LogFCs of individual stress time points vs. controls in both hemispheres and significant genes as rows was used as input. K-means clustering was performed using the kmeans function R "stats" package. We used the R package "NbClust" to determine the appropriate number of centers, however no consensus was found between different rating methods. Thus, we opted to use a manual approach where the number of centers was increased until new clusters did not contain vastly different expression profiles. The number of clusters was different between phosphoproteomic and transcriptomic datasets due to the difference in number of time-points and the difference in complexity of expression profiles. Using this approach, we found 5 clusters to be most appropriate for the phosphoproteomic and 25 clusters to be most appropriate for the transcriptomic dataset. This approach may be subject to human bias.

**Tissue processing and immunohistochemistry**. In order to confirm the localization of fluorescent tags in excitatory neurons in the CaMKIIa-nuTRAp mice, in inhibitory neurons in the vGAT::bacTRAP mice, and in both neurons and glia in CMV-nuTRAP mice we performed immunohistochemistry as previously described[121]. Briefly, the mice were first deeply anesthetized with pentobarbital (150 mg/kg, i.p.) and perfused intracardially through the left ventricle for 2 mins, with approximately 20 mL of artificial cerebrospinal fluid (ACSF) or ice-cold PBS pH 7.4. Then the brain was dissected, blocked and fixed for 1.5-3 hours in ice-cold paraformaldehyde solution (4% PFA in PBS, pH 7.4). The tissue was cryo-protected, frozen in tissue mounting medium (Tissue-Tek O.C.T Compound, Sakura Finetek Europe B.V., Netherlands), and sectioned coronally into 40 μm thick sections.

The brain sections were incubated in primary antibody solution containing 0.2% Triton X-100, and 2% normal goat serum in PBS, at 4 °C under continuous agitation, overnight. After 3 washes with PBS the sections were transferred in secondary antibody solution with 2% normal goat serum in PBS. Then the sections were washed again in PBS, mounted onto glass slides (Menzel-Gläser SUPERFROST PLUS, Thermo Scientific), air-dried and coverslipped with Dako fluorescence mounting medium (Agilent Technologies). For anti-GAD65/67, an additional step of antigen retrieval and one blocking step preceded the primary antibody incubation. During antigen retrieval, the sections were submerged in 500 μL/well citrate buffer 10 mM, pH 6, containing 0.05% Tween20 and remained there for 20 mins, under soft agitation (30 rpm). Then the sections were rinsed with PBS and incubated in 500 μL/well blocking solution (5% NGS, 0.3% TritonX-100 in PBS), under soft agitation, at RT for 60 mins. The primary antibody solution contained 0.3% Triton X-100, and 5% normal goat serum, and the

secondary antibody solution contained 0.05% Triton X-100, and 5% normal goat serum in PBS.

Primary antibodies used: chicken anti-GFP (ab13970, Abcam, 1:1000), chicken anti-GFP (GFP-1020, Aves, 1:5000), rabbit anti-mCherry (ab167453, Abcam, 1:1000), rabbit anti-GAD65/67 (AB1511, Millipore, 1:500), rabbit anti-NeuN (EPR12763, ab177487, Abcam, 1:300), mouse anti-GFAP (G3893, Sigma-Aldrich, 1:400). Secondary antibodies used: goat anti-chicken Alexa Fluor 488 (A-11039, Thermo Fischer Scientific, 1:1000), goat anti-rabbit Alexa 546 (A11035, Thermo Fisher Scientific, 1:300), goat anti-mouse Cy™3 (115-165-003, Jackson ImmunoResearch, 1:300), goat anti-rabbit Alexa Fluor Plus 647 (A32733, Invitrogen, 1:500). Stains used: Nissl stain (N21483, NeuroTrace 640/660 Nissl stain, Thermo Fischer Scientific, 1:300), DAPI (D3571, Thermo Fischer Scientific, 1:2000). Microscopy images were acquired in a confocal laser-scanning microscope (CLSM 880, Carl Zeiss AG, Germany), maintaining a pinhole aperture of 1.0 Airy Unit and image size 1024x1024 pixels. Images were acquired using a Z stack with a 20x objective and pixel size 0.59 mm.

**Western blot**. Tissue was mechanically homogenized in 200 μL TEAB lysis buffer and sonicated for 2 mins at maximum intensity. Following centrifugation at 16000 g at 4 °C for 30 mins, the recovered supernatant was collected and the protein concentration was determined using the Qubit Protein Assay Kit (Q33211, Thermo Fischer Scientific). Proteins were denatured by boiling at 95 °C in SDS-loading buffer containing 10% mercaptoethanol, for 5 mins. 20 μg of total protein/sample were loaded in each well of a 10% PAGE gel and transferred onto a nitrocellulose membrane (#1704158, BioRad) in a Trans-Blot Turbo system (Bio-Rad). Membranes were blocked in a 5% non-fat milk solution in 0.1% Tween20 (TBS-T), for 1 hour at RT and incubated in primary antibody solution overnight at 4 °C, or 2 h at RT. Subsequently, membranes were washed three times in TBS-T, incubated in secondary antibody solution at RT for 1 h, washed again three times in TBS-T and visualized in a ChemiDoc Imaging System (Bio-Rad). Antibodies were diluted in antibody solution containing 1% non-fat milk in TBS-T. Primary antibodies used are the anti-pS553-Syn1 rabbit monoclonal (ab32532, Abcam, 1:10000) and anti-GAPDH rabbit polyclonal (ABS16, Millipore, 1:1000). The primary antibodies were targeted with the secondary antibody goat anti-rabbit IRDye800CW (926-32211, LI-COR Biosciences GmbH, 1:10000).

**RT-PCR**. Real-Time polymerase chain reaction (RT-PCR) was performed using SYBR green (04887352001, Roche) on a CFX384 Touch Real-Time PCR Detection System (Bio-Rad). As described before[23], the cycling protocol was: 5 mins at 95 °C, then 45 cycles each including a step of denaturation (10 s at 95 °C), a step of annealing (10 s at 60 °C), and a step of elongation (10 s at 72 °C). Primers were designed with Quantprime[122] and tested for quality and specificity by melt-curve analysis and gel electrophoresis, unless stated otherwise. The primers for AtLAS were as previously published[123].

Forward (FP) and reverse (RP) primer sequences:

*Gad1* FP: GGTCCTCTTCACCTCAGAACACAG, RP: TTGTCGGTTCCAAAGCCAAGCG

*Slc6a1* FP: GTGTTGGTTGGACTGGAAAGGTG, RP: AAGCGTCACTCCACGGAAGAAC

*Gfap* FP: TGGCCACCAGTAACATGCAAGAG, RP: CGTCTGTGAGGTCTGCAAACTTAG

*Adora1* FP: TGGCTCTGCTTGCTATTGCTGTG, RP: TGAGTCACCACTGTCTTGTACC

*Neuro6d* FP: ATCTGCGCAGCCAATCTCTCAC, RP: TGCCAATTACGCAGCCCACAAG

*Nrn1* FP: TGATCCTCGCGGTGCAAATAGC, RP: AAGCCCTTAAAGACTGCATCACAC

*Slc17a7* FP: GTCCATGGTCAACAACAGCACAAC, RP: AGTTGAACTGGGCTTTCTGCAC

*AtLAS* (AK013786) FP: ACAGATGGCAAGATGAGG, RP: GCCTTTGACCTCTTTGG

*Tubd1* FP: TCTCTTGCTAACTTGGTGGTCCTC, RP: GCTGGTGTCTTTAAATCCCTCTACG

*Hprt* PF: GTTGGGGCTTACCTCACTGCTTTC, PR: CCTGGTTCATCATCGCTAATCACG

**Statistics & reproducibility**. We used a block design for experiments. Animals and samples were split into multiple blocks, containing one replicate of each condition. Experimental and processing order within these blocks was randomized. Investigators were blinded during behavior and sample processing, but not during the analysis process. However, the same algorithmic analysis methods were used for all samples within each experiment. No statistical method was used to pre-determine sample size. No data were excluded from the analyses.

**Reporting summary**. Further information on research design is available in the Nature Research Reporting Summary linked to this article.

## Data availability

The LC-MS/MS data generated in this study have been deposited in the ProteomeXchange database under accession codes "PXD024829" and "PXD029903".

The sequencing data data generated in this study have been deposited in the Gene Expression Omnibus database under accession codes "GSE169505" (bulk and TRAP sequencing), "GSE169509" (smallRNA sequencing) and "GSE169510" (snRNA sequencing) Selected statistical results showing group contrasts and gene/protein clusters are reported in Supplementary Data 1 (phosphoproteomics), Supplementary Data 2 (transcriptomics) and Supplementary Data 3 (proteomics). The interactive web app can be accessed at https://bohaceklab.hest.ethz.ch/StressomeExplorer Source data are provided with this paper.

## Code availability

Code for the interactive web app and all analyses (independent scripts) can be found at https://github.com/ETHZ-INS/StressomeExplorer, https://doi.org/10.5281/zenodo.6033982

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

## Acknowledgements

The lab of JB is funded by the ETH Zurich, the ETH Project Grant ETH-20 19-1, SNSF Grant 310030_172889, 3R Competence Center, Kurt und Senta Herrmann-Stiftung, the Novartis Foundation for Medical-Biological Research and the Swiss Foundation for Excellence and Talent in Biomedical Research. This work was also supported by ETH Research Grant ETH-25 20-2 to PLG, SNSF Grant 197888 to HUZ, European Research Council Starting Grant (803491, BRITE) to FvM, and the Botnar Research Centre for Child Health, Multi-Investigator Project to JB and FvM. We thank the staff of the EPIC for the excellent animal care and their service to our animal facility. We thank Prof. Markus Stoffel for generously providing CMV-Cre mice. Parts of figures were created using Biorender.com.

## Author contributions

L.v.Z.: conceived experiments, conducted experiments & tissue processing (phospho-proteome immediate & time-course, time-course transcriptomics, proteome), data analysis, graphs & figures, interpreted results, wrote manuscript; A.F.S.: conceived experiments, conducted experiments & tissue processing (TRAP), graphs & figures, interpreted results, wrote manuscript; R.W.: assistance with TRAP sample processing, conducted single-nucleus RNAseq; R.R.D.G.: helped establish TRAP protocol, assistance in TRAP sample processing; OS: conducted experiments (behavior, assistance with time-course transcriptomics & immediate phosphoproteomics); K.G.: conducted small-RNAseq; C.A.M.: produced western blots; T.K.: assisted with the MS; H.Y.L.: assisted with sample processing; S.N.D.: conducted experiments (assistance with time-course phosphoproteomics, behavior); M.P.: conducted experiments (behavior); L.H.: helped establish the snRNAseq protocol; assistance with snRNAseq. H.U.Z.: lab space and resources for 4 rounds of TRAP sample processing, advice; F.v.M.: lab space, expertise and resources for snRNAseq, advice; P.L.G.: conducted data analysis and data integration, generated graphs & figures, interpreted results, wrote manuscript; J.B.: conceived experiments, interpreted results, provided resources and funding, wrote manuscript.

## Competing interests

The authors declare no competing interests.
