## [Peer Review File · Nature Communications]

Reviewers' Comments:

Reviewer #1:

Remarks to the Author:

In their article Molecular roadmap of the healthy stress response in the mouse hippocampus, von Ziegler and colleagues present an extensive array of *omic experiments designed to probe molecular effects of acute stress (AS). While the amount of data and analyses are impressive, and the individual assays appear to have been thoughtfully deployed based on the expected time-scales of the events following AS, one cannot help feel an opportunity has been missed in probing the combination or joint interpretation of these data, or at least to highlight more clearly specific regulatory events/pathways with relevance to AS. Beyond the lack of an overarching theme/result I have the following concerns:

I wasn't able to access any supplementary tables containing normalised data or differential expression results from any of these experiments. In the absence of overarching results/themes it would presumably be of great benefit to the community to provide these as a resource for other groups to further analyse. Equally, the plots generated on the web application are static images - while this is fine it would be good to have a facility to download the relevant data behind these plots, either in bulk, or per-gene.

Generally the authors rely heavily on k-means clustering however there is no mention anywhere in the Results or Methods how the choice of k was made. Given that it features heavily in the interpretation of the first 3 figures of their manuscript it would be of great benefit to be provided more insight here. Usually analyses using k-means will employ a less biased (compared to human intuition) statistically driven approach to cluster number selection, for example the gap statistic.

It is not entirely clear what is meant by 'behavioral clustering cross validated with t-sne' or how k was selected for k-means. T-sne 'cross validation' is a little misleading as only 5% of the data are used - this is really more of a QC/test-set than a cross validation. Further, I'm not convinced t-sne clustering adds anything here (beyond a colourful picture), in my opinion it does not lend a great deal of support to the k-means clusters - it would likely agree with any arbitrary choice of k. Instead of devoting figure real-estate to the t-sne, perhaps some graphical representation of the fraction of behavioural traits assigned to or enriched in each cluster would be more helpful to the reader. Also, I was unable to find any description of how these traits were 'visually inspected' and assigned to clusters.

Given the major advantage, acknowledged in the Discussion, of the TRAP data is the cell type-specificity rather than translational dynamics (more in the realm of ribosome footprinting) it feels as though the authors missed a potential opportunity to jointly analyse (or at least jointly interpret) snRNA-seq and TRAP data from the relevant neuronal sub-types. That said, given the differentially expressed genes in TRAP, one wonders whether restricting the proteomic analysis to these genes would yield some potentially confirmatory result? Can the snRNA-seq reveal genes differentially expressed in TRAP that do not have confounding expression in off-target cell-types? Are these genes (if any) more likely to be seen at the protein level by mass-spec?

Finally, several figures suffer from very low-resolution images, some specific examples noted in minor comments below

Minor Comments

- First use of HC on page 6 could be made more clear "dorsal and ventral hippocampus (dHC and vHC)"
- "Immediately" after stress - presumably some time delay to sacrifice?
- Resolution of most parts of Fig 2 quite poor
- Volcano plots in Fig 3B may benefit from adding a ceiling to the p-value axis
- FC scatter plots in Fig 3E/F could be labelled more clearly. A fold-change implies a difference but the axes only show one label. Are all these relative to Control? E.g is the first plot $\log(45\text{min}/\text{Control})$ for dHC (x-axis) and vHC (y-axis)? Not clear from the legend - text in the Results section mentions 'consecutive timepoints'
- Again - in Fig 4C, logFC relative to what? Control? Reversing the order of processed and

unprocessed throughout Fig 4 may make sense given the latter precedes the former...

- Most of Fig 5 suffers low resolution images, again in C) the logFC is unclear what is being compared, and E) is missing a colour legend
- Not clear why two different differential expression packages (EdgeR and DESeq2) were used for normalisation in the bulk+smRNA vs TRAP analyses, respectively. DESeq2 is also not mentioned in the Software and Code section of the Reporting Summary
- Fig 6K is both very low resolution and an extremely inefficient way to visualise these genes
- The use of the word 'increasing' could be misleading in the sentence 'Only by increasing the significance cutoff to a FDR of 0.5' - use 'relaxing' instead?

Reviewer #2:

Remarks to the Author:

Ziegler et al. comprehensively profile the molecular mechanisms underlying the acute stress response using multiomic profiling of the dorsal and ventral hippocampus. The authors present an impressive amount of data, and provide a detailed picture of the healthy stress response across time, and a variety of molecular scales. Particularly noteworthy are the efforts to address effects both upstream and downstream of transcription, exploring the whole cascade of events from start to finish and addressing commonly less studied elements, such as phosphorylation and the translome. The tremendous amount of data coupled with the open science efforts of the authors make this study a very valuable resource for the scientific community. The extensive analyses are robust, rigorous and elegant and experiments are appropriately powered and well-designed. My considerable enthusiasm for this work is somewhat tempered by the limited integration across levels of analysis that could deliver greater biological insight. I have a number of suggestions to increase the impact of this work.

Major Points

- Overall, this is missing a more in-depth approach to integrating the different levels of analysis. The structure of the discussion in particular could be improved to facilitate understanding of how the different levels of analysis connect to each other. Are there conclusions that can be drawn on a biological level when integrating several levels of analysis? Intuitive examples would be bulk and single-cell RNA sequencing; or is there a possibility to trace the molecular cascade all the way from phosphorylation to the proteome? Such an integration would tremendously enhance the impact of the paper.
- The approach to integrating mice of different sexes seems haphazard and inconsistent. The authors should explain their approach to integrating sex as a biological variable and the implications of their approach. Most experiments are done exclusively in males with the exception of a bulk RNA-sequencing experiment in males and females reported in the supplemental and a subset of TRAP-seq experiments conducted exclusively in females. The lack of differences in bulk-RNA seq is an interesting result to report (although I note there are small and unequal sample sizes that may limit the ability to detect significant differences). However, it is not appropriate to extrapolate from this a general lack of sex differences across other untested levels of analysis. Given that an overarching conclusion of this work appears to be that the phospho-proteome is most relevant to understanding the acute stress response, it would have been interesting to see the sex comparison in this experiment.
 - o Given the inconsistencies in sex, the authors should pay attention to reporting the sex of the animals for each experiment (e.g., proteome).
 - o The title should be amended to indicate that this is in the male mouse.
- The detailed behavioral analysis is methodologically sound but feels out of context. The in-depth display of specific behavioral effects of forced swim stress do not add to the multi-omic data, especially because any specific behavior of this separate cohort of animals cannot be directly related to any of the molecular findings. Ultimately, the machine learning analysis results do not alter the interpretations based on standard behavioral testing. Further, the most interesting time-points to apply a sensitive behavioral assessment which could be expected to reveal different findings (i.e. intermediate time-points) are not included. I would recommend significantly shortening this part or moving substantial portions to the supplementary material.
 - o Multiple testing corrections in the behavioral data are not reported. When comparing behaviors (here clusters) between the control and stressed group, the multiple testing correction should take

into account the number of comparisons made. It is unclear whether the p-values were adjusted.

- The claim that 'no lasting alterations in anxiety-related behavior' are observed is overstated and not fully supported by the data. Both the conventional and ML analysis reveal effects at 24h post stress: increased grooming and decreased supported rearing, both of which may be interpreted as indicative of a lasting alteration in anxiety-like behavior. While this could be seen as a minor point given the lack of differences in other behavioral measures, it is worth noting given that it is at odds with the tight temporally-resolved molecular cascades reported here. The claims to complete lack of differences should be tempered and some brief discussion would be of interest.

- The snRNA-seq data is underutilized and the reporting of this experiment is difficult to follow. Was a statistical comparison of differential expression analysis conducted on this data? It seems that lists of DEG genes identified in bulk RNA-seq were cross-referenced to FC values from snRNA-seq to attribute DEGs to specific cell types. Why not simply analyse DE in the snRNA-seq data within each cell-type? It could then be interesting to compare these results to bulk RNA-seq to examine potential convergence.

Minor Points

- I have questions concerning a number of choices made by the authors related to the study design:

- o The authors chose the forced swim test (FST) as an acute stressor. The authors should clearly articulate that some findings could be specific to the FST and not necessarily apply to other stressors.

- o The authors report that mice were single housed 24h prior to performing the FST. However, it is not clear why this choice was made. This is somewhat concerning, given that single housing is a stressor in itself, in particular for female mice. The authors should justify this choice (e.g., by presenting pilot data or from available literature).

- o It is unclear why a given series of time points was chosen for each level of analysis (e.g., transcriptomic analysis). Were there pilot studies to determine the optimal time points? Or was this based on time points reported in the literature?

- o Why cut the hippocampus into two halves rather than thirds? The referenced paper from Fanselow & Dong identifies three divisions along the dorsoventral axis: dorsal, intermediate and ventral. Could inclusion of intermediate hippocampus in dorsal and ventral contribute to observed correlations?

- Some of the plots are hard to read because the authors try keeping identical axis limits across different subplots (e.g., Fig 3, B E F). While this is generally a good approach, it becomes limiting when the data points are shrunken such that differences between plots can no longer be appreciated. I would suggest flexible axis labeling in cases where it would enhance readability of the plot for the readability.

- Group ordering in some plots is not intuitive. For example 4A presents processed to the left of unprocessed. This is a minor point but reversing order such that unprocessed precedes processed would make interpretation of claims relating to temporal sequence of events easier to visually inspect in the figures.

- For any k-means clustering, it is not reported how the number of initial centers was chosen. This can be influential on the outcome if the number of centers initially chosen is too small.

- I would refrain from reporting on results with 50% FDR, as this indicates that half of what is reported may be false (see Results/Proteome). I would suggest removing these results from the text.

- The authors should limit descriptions to observable behavioral states. For example, rather than describing mice as 'anxious', discuss the relevant behavioral metrics that indicate anxiety-like behavior.

- Reporting of results would benefit from more specific conclusions. For example, a detailed analysis of SYN1 phosphorylation is presented but it is unclear what we should conclude from the findings.

Reviewer #3:

Remarks to the Author:

von Ziegler and colleagues examined the molecular cascades that occur in mouse hippocampus following a brief forced swim stressor. The authors broadly found that gene expression, protein phosphorylation, and protein translation events that occur in response to acute stress resolve

within four hours of the stressor, which parallels the brief timescale of behavioral response to the stressor. The authors describe this acute stress response as adaptive homeostasis. This study is particularly valuable as the authors investigate multiomic molecular response across a time course, and because of the interactive web portal authors have developed to public browsing of their datasets. There is a minor attempt to understand sex differences in acute stress response. Analysis of hippocampal subregion specificity is useful given previous evidence on separate dorsal/ventral and CA1/3/DG functions in stress response, and it is also of interest that the authors discovered correlation between expression changes at early time points, indicating similarity in early stress response. The analyses examining correlations of change between consecutive time points to discover that expression/protein changes are not random but unfold over time are particularly elegant. Cell-type specificity of transcriptomic change at the 45m time point is also made, in male (?) whole-hippocampus. This could easily be several papers, but is nevertheless useful as one complete piece. Overall, the science is extremely well-done and this manuscript is well-organized and well-written. I have only minor comments to clarify points for this exciting body of work.

Minor Comments:

1. The overall conclusions from the DLC-based behavioral characterization presented in Figure 1 are very useful, although the conclusions are not different from those garnered from the gross-scale behavioral characterization (S1). The main difference is that the DLC-based analysis is very complicated and non-intuitive. In particular, the correlation analysis does not lead to any conclusions useful for understanding resolution of the stress response. This figure would be more useful if the authors overlaid information given in the text on specific types of behaviors post-hoc associated with the behavior clusters (rears, walking, grooming). Perhaps a more useful correlation would be a 4h vs 24h comparison.
2. It is unclear in many of the heatmaps how the data is sorted along the y-axes (the types and time bins make sense). For example, in 4A and 4D, it appears vaguely but not-quite sorted by expression change at 45min. Sorting in 5D-E are also unclear. Can the authors clarify how genes are ordered in the figure legends?
3. It is interesting that the snRNA-seq generally did not show expression differences between Ctl and AS samples within any cell type. The authors have done a nice job using that data to understand the bulk RNA-seq expression differences, but additional discussion of the lack of snRNAseq differences is warranted.
4. Why would statistical power of TRAP-seq be lower than bulk-RNAseq, given the higher specificity of TRAP-seq for active changes? The authors have attempted to get around a lack of significant alterations by re-combining with bulk-seq data, but the justification for this is insufficient. The rest of the manuscript is so thorough, it does not need the less-conclusive TRAP data, despite all of the work that surely went into these experiments. The cellular specificity of some of the IEGs is interesting, but surely the snRNAseq data should point to the same conclusions?

General Response to all reviewers

We thank all the reviewers for the very positive evaluation of our work and we appreciate the constructive suggestions on how to improve the manuscript. We provide detailed responses below, but first we address a general issue raised by all reviewers, the suggestion that we should try to better link the different levels of our multi-omic analyses. This is indeed a very challenging request for two reasons: On the one hand, different levels of analysis have different technical constraints (e.g. transcriptomics is very sensitive and samples all genes, proteomics samples only a subset of proteins and transcription factors and nuclear proteins are notoriously difficult to detect). On the other hand, the different molecular processes have different temporal dynamics, so it can be very difficult to find the "right" time points that capture any given gene/target across all pathways (from phosphorylation to active transcription to bulk transcriptome to the ribosome and then to the finished protein product). That being said, we have now included many new analyses to better integrate our data across scales and to also connect and cross-reference it more extensively with existing datasets.

1. We identified the MAPK/ERK-CREB pathways, which had already been highlighted in the phosphoproteomic data in the original manuscript, as a key stress-induced process that plays out across molecular levels. First, we cross-referenced our transcriptomic results against published cell culture data (Tyssowski et al, Neuron, 2018) and find that the transcriptomic response after acute stress fully recapitulates the immediate MAPK/ERK-dependent changes observed in neuronal cell culture (Figure S4D). Second, we took this set of rapid-response genes and overlaid it with the single-nucleus RNAseq data. Surprisingly, this showed that these rapid response genes often turn on in several different cell types, and that their reactivity is independent of their baseline expression level (Figure 5C). Third, we searched for stress-responsive MAPK/ERK-CREB target genes on the level of the translome and found that members of this pathway are indeed over-represented in actively translated genes after stress exposure. Thus, we identify a thread that can be followed throughout the entire multi-level molecular cascade. We now dedicate a whole section in the discussion to these observations, and also discuss the challenges inherent in linking data across molecular levels (section: "Trends across the multi-omic molecular response").
2. As part of the effort to better cross-link different levels of analysis, we compared the ubiquitous CMV-TRAP analysis (translation) with the bulk-RNAseq analysis (transcription). Interestingly, we observed that only the largest changes on the mRNA level led to changes on the translational level (Figure 6H). This is reminiscent of translational buffering as observed in a recent report that used a very strong manipulation (kainate-induced sustained seizures, Fernandez-Albert et al, Nat. Neurosci, 2019). We compared our dataset to theirs and observed that the same top-regulated genes were increased in translation in both models (Figure S7E). This strengthens our initial suspicion that stress-induced molecular changes are buffered at the level of translation. We address this in the discussion.
3. To further explore the regulatory events controlling the observed gene expression changes, we tried to perform TF activity analysis (as was originally done in bulk, Figure 3D) on the snRNAseq data. However, due to the low sample size (n=2/group), the lack of cell-type-specific targets, and the indirect nature of activity inference, we were not sufficiently confident about the robustness of the results to include them in the paper. Instead, we linked the bulk TF analysis to snRNAseq by describing the relative expression of the identified factors in the different cell types (Figure 3D, right panel). This revealed different sets of neuronal and non-neuronal factors, which can serve as hypotheses for future research.
4. We deepened our snRNAseq analysis as suggested by several reviewers. First we clarified that already in the original manuscript we had performed a differential gene expression analysis on the snRNAseq data and identified many significant changes. Second, we added statistical analyses of the snRNAseq data to the online Stressome App, which allows users to interactively search and download all our data and results. Third, we performed pathway analyses on the snRNAseq data and found very different pathway enrichments in neurons, astrocytes and

vascular cells (Figure 5B). All these results were integrated into the main manuscript, so that Figure 4 was expanded into new Figures 4 and 5.

5. We compared the neuronal subset of snRNAseq with the TRAP datasets. As expected, given the different temporalities they reflect, there was limited correlation between the two types of data. Indeed, our data suggest that for many genes with increased translation in neurons after acute stress, the corresponding transcriptional burst is over at the 45 min time point. Nevertheless, we identified a small set of IEGs that show a reproducible response across these datasets, representing those stress-induced components, which are in the right time-window to be captured by both assays (see response to Reviewer 1, Section 1.4, Figure R2).
6. We tried to stratify the gene expression profiles from the bulk sequencing (Figure 2E) using our single-cell data to try to see if any of these profiles are enriched for genes expressed in certain cell types (Figure R1). However, we did not find clear overlaps, and different enrichment methods (ORA, GSEA and GSVA) did not converge in their conclusions. This is due to the fact that we were only able to investigate if genes with higher baseline expression in certain cell types were also enriched in profiles from the bulk sequencing data. However, as we noticed in Figure 5C, baseline expression and stress responsiveness of genes within cell types does not necessarily correlate. In order to properly address this issue, a time-series experiment using snRNAseq would be required. While this is beyond the scope of the current manuscript, this is an important insight for future work, which we have included in the discussion section.

Figure R1: Enrichment of bulk transcriptional clusters (from Figure 2E) in the broad cell types identified with snRNAseq. The right heatmap shows the clusters' temporal profile. The left heatmap shows the overlap coefficients between the clusters and the cell types with the highest average expression of the respective genes. The middle heatmap shows the proportion of cell type signatures in each of the clusters.

REVIEWER COMMENTS

Reviewer #1 (Remarks to the Author):

In their article Molecular roadmap of the healthy stress response in the mouse hippocampus, von Ziegler and colleagues present an extensive array of *omic experiments designed to probe molecular effects of acute stress (AS). While the amount of data and analyses are impressive, and the individual assays appear to have been thoughtfully deployed based on the expected time-scales of the events following AS, one cannot help feel an opportunity has been missed in probing the combination or joint interpretation of these data, or at least to highlight more clearly specific regulatory events/pathways with relevance to AS. Beyond the lack of an overarching theme/result I have the following concerns:

RE: We thank the reviewer for the positive assessment of the quality and breadth of our analyses. Regarding the interpretation of our data across molecular scales, we have addressed this point in detail in our general response to all reviewers (pages 1-2 above).

1.1. I wasn't able to access any supplementary tables containing normalised data or differential expression results from any of these experiments. In the absence of overarching results/themes it would presumably be of great benefit to the community to provide these as a resource for other groups to further analyse. Equally, the plots generated on the web application are static images - while this is fine it would be good to have a facility to download the relevant data behind these plots, either in bulk, or per-gene.

RE: All the normalized data and DEA results were available in the github repository using standardized Bioconductor formats. In response to the reviewer's comment, we have improved the repository's documentation, and additionally made the data available through our online Stressome App. A new tab now enables the user to download all data associated with a selected gene as an excel file with multiple sheets, one for each dataset. This tab now also links to the relevant github directories with all analysis results and the complete processed data.

<https://github.com/ETHZ-INS/StressomeExplorer/tree/main/Results> and <https://github.com/ETHZ-INS/StressomeExplorer/tree/main/data>

1.2. Generally the authors rely heavily on k-means clustering however there is no mention anywhere in the Results or Methods how the choice of k was made. Given that it features heavily in the interpretation of the first 3 figures of their manuscript it would be of great benefit to be provided more insight here. Usually analyses using k-means will employ a less biased (compared to human intuition) statistically driven approach to cluster number selection, for example the gap statistic.

RE: Thank you for pointing out this omission. We indeed looked at statistical methods for cluster number selection, such as the "NbClust" package for R (Charrad et al, 2014: <http://www.jstatsoft.org/v61/i06/>), which uses a combination of 30 different methods (including popular methods such as the elbow, the silhouette and the gap statistics method) to suggest the best number of centers by consensus. However, for our data this method yielded a very low number of centers (the most popular number of centers across all methods was 2, with some methods also suggesting between 10-25 centers). Two centers made little sense to us and clustered genes with completely different expression profiles together. Therefore, for the choice of k in Figures 2 and 3 we settled on a manual approach that yielded most distinctive expressional profiles. Once the increase in k yielded only more clusters with similar profiles but slightly different magnitudes, we stopped increasing k. We added a section to the methods to clarify this (section: "K-means clustering"). As explained in the next comment, we removed the unsupervised behavior profiling and the associated clustering from Figure 1.

1.3. It is not entirely clear what is meant by 'behavioral clustering cross validated with t-sne' or how k was selected for k-means. T-sne 'cross validation' is a little misleading as only 5% of the data are used

- this is really more of a QC/test-set than a cross validation. Further, I'm not convinced t-sne clustering adds anything here (beyond a colourful picture), in my opinion it does not lend a great deal of support to the k-means clusters - it would likely agree with any arbitrary choice of k. Instead of devoting figure real-estate to the t-sne, perhaps some graphical representation of the fraction of behavioural traits assigned to or enriched in each cluster would be more helpful to the reader. Also, I was unable to find any description of how these traits were 'visually inspected' and assigned to clusters.

RE: All reviewers had criticised that the in-depth, unsupervised behavior analysis added little to the main message of the paper and distracted from the molecular focus of our work. We agree with this assessment and we followed the suggestion of Reviewer 2 to significantly shorten the behavior part. We thus removed the unsupervised analysis including clustering from the manuscript and moved the remaining behavior analyses - which show that most of the stress-induced behavioral changes in the open field and elevated plus maze are very short-lasting - into the supplementary section (Figure S1).

1.4. Given the major advantage, acknowledged in the Discussion, of the TRAP data is the cell type-specificity rather than translational dynamics (more in the realm of ribosome footprinting) it feels as though the authors missed a potential opportunity to jointly analyse (or at least jointly interpret) snRNA-seq and TRAP data from the relevant neuronal sub-types. That said, given the differentially expressed genes in TRAP, one wonders whether restricting the proteomic analysis to these genes would yield some potentially confirmatory result? Can the snRNA-seq reveal genes differentially expressed in TRAP that do not have confounding expression in off-target cell-types? Are these genes (if any) more likely to be seen at the protein level by mass-spec?

RE: We agree that the cell-type specificity is one of the major advantages of TRAP-seq, but it is also the most sensitive technique to report which mRNAs actually become translated in a given cell type. However, due to the highly dynamic nature of the stress-induced transcriptional response, TRAP (which captures mature and translated mRNAs that are relatively long-lived) does not recapitulate the snRNAseq (which, because it chiefly captures unprocessed transcripts, provides a much more transient snapshot of active transcription). This is already seen when comparing bulk RNAseq to snRNAseq: While we obtained good fold-change correlations between the snRNAseq and the unspliced fraction of the bulk RNAseq (Figure 4D), the correlation was considerably lower with the spliced fraction. This is because the very first wave of the transcriptional response, while having disappeared from active transcription at 45 min, remains visible much longer in the processed transcriptome. Furthermore, the fact that ubiquitous (i.e. not cell-type specific) TRAP does not recapitulate a large part of the transcriptome either, indicates that the key difference is not cell type specificity, but the enrichment for translated transcripts.

We nevertheless directly compared the neuronal subset of snRNAseq with the TRAP datasets, as shown below. As expected, given the different temporal processes they reflect, there was limited correlation between the two types of data. Indeed, our evidence suggests that for many genes with increased translation in neurons after acute stress, the corresponding transcriptional burst is over at the 45 min time point. Nevertheless, we identified a small set of IEGs showing a reproducible response across these datasets, representing those translated components of the response, which are in the right time-window to be captured by both assays (Figure R2).

As suggested by the reviewer, we also restricted the proteomic analysis to stress responsive genes in the TRAP data. Out of 50 TRAP-stress genes, only 8 could be quantified at the protein level (one of the technical limitations inherent with current state-of-the-art proteomics). However, we did not find any significant changes in these 8 proteins, despite the fact that we had more power due to reduced multiple testing corrections. It is likely that we would need to generate more fine-grained time-course datasets for both TRAP-seq and proteomics to better reconstruct the cascades from transcription to protein expression. However, these costly experiments would be beyond the scope of this work. We decided not to include this data in the manuscript, but we touch on the technical limitations in a new section in the discussion: "Trends across the multi-omic molecular response".

Figure R2: Log-fold change comparison upon stress between snRNA and TRAP data in excitatory and inhibitory neurons.

1.5. Finally, several figures suffer from very low-resolution images, some specific examples noted in minor comments below

RE: We apologize for this inconvenience, the low resolution resulted from a limit on the maximum file size in the original submission. We have now uploaded high-resolution, publication-quality figures.

Minor Comments

1.6. First use of HC on page 6 could be made more clear “dorsal and ventral hippocampus (dHC and vHC)”

RE: Thank you, we implemented this change.

1.7. "Immediately" after stress - presumably some time delay to sacrifice?

RE: Mice were euthanised within a maximum of 1 min after stress. The dissection procedure took approximately 4 min and was performed at 0°C using PBS ice. We now clarified this in the method section “Swim stress paradigm and tissue collection”).

1.8. Resolution of most parts of Fig 2 quite poor

RE: Higher resolution figures have been uploaded

1.9. Volcano plots in Fig 3B may benefit from adding a ceiling to the p-value axis

RE: We thank the reviewer for this helpful suggestion, we have updated the figure accordingly by setting a ceiling p-value of 1e-20 (now Figure 2B).

1.10. FC scatter plots in Fig 3E/F could be labelled more clearly. A fold-change implies a difference but the axes only show one label. Are all these relative to Control? E.g is the first plot log(45min/Control) for dHC (x-axis) and vHC (y-axis)? Not clear from the legend - text in the Results section mentions ‘consecutive time points’

RE: Because we have added new data, these graphs are now in Figure S3D/E. Indeed, the first plot is [logFC of stress(45min) vs control in vHC] vs. [logFC of stress(45min) vs control in dHC]. Thus, the axis “logFC dHC” should rather read “logFC ctrl vs stress in dHC”. Due to space constraints, we changed the legend to clarify this point.

1.11. Again - in Fig 4C, logFC relative to what? Control? Reversing the order of processed and unprocessed throughout Fig 4 may make sense given the latter precedes the former...

RE: Thank you for these sensible suggestions, we changed the order of "processed" and "unprocessed" plots accordingly (now Figure 3A/B). We also changed "mean logFC" to "mean logFC vs control" (now Figure 3C).

1.12. Most of Fig 5 suffers low resolution images, again in C) the logFC is unclear what is being compared, and E) is missing a colour legend

Re: We added higher resolution figures, clarified what logFC means (swim vs. control) (now Figure 4D) and added the missing colour legend (now Figure 5A). Thank you for pointing out these details.

1.13. Not clear why two different differential expression packages (EdgeR and DESeq2) were used for normalisation in the bulk+smRNA vs TRAP analyses, respectively. DESeq2 is also not mentioned in the Software and Code section of the Reporting Summary

RE: DESeq2 was not used for differential expression analysis. Only its variance-stabilizing transformation (something edgeR doesn't have) was used, in order for the input to SVA analysis to abide by the assumptions of the method.

1.14. Fig 6K is both very low resolution and an extremely inefficient way to visualise these genes

RE: The low resolution issues have been solved. However, we would argue that the polar plots are a rather useful way to present the differences in gene expression profiles between cell types (and bulk), particularly because of the low number of genes. Given that different readers respond to different modes of visualization, we hope the reviewer accepts that we would prefer to keep this figure.

1.15. The use of the word 'increasing' could be misleading in the sentence 'Only by increasing the significance cutoff to a FDR of 0.5' - use 'relaxing' instead?

RE: We changed the wording to "...we relaxed the significance cutoff to a FDR of 0.5" (page 10).

Reviewer #2 (Remarks to the Author):

Ziegler et al. comprehensively profile the molecular mechanisms underlying the acute stress response using multiomic profiling of the dorsal and ventral hippocampus. The authors present an impressive amount of data, and provide a detailed picture of the healthy stress response across time, and a variety of molecular scales. Particularly noteworthy are the efforts to address effects both upstream and downstream of transcription, exploring the whole cascade of events from start to finish and addressing commonly less studied elements, such as phosphorylation and the translome. The tremendous amount of data coupled with the open science efforts of the authors make this study a very valuable resource for the scientific community. The extensive analyses are robust, rigorous and elegant and experiments are appropriately powered and well-designed. My considerable enthusiasm for this work is somewhat tempered by the limited integration across levels of analysis that could deliver greater biological insight. I have a number of suggestions to increase the impact of this work.

RE: We appreciate the reviewers positive evaluation of our work and the constructive feedback.

Major Points

2.1. Overall, this is missing a more in-depth approach to integrating the different levels of analysis. The structure of the discussion in particular could be improved to facilitate understanding of how the different levels of analysis connect to each other. Are there conclusions that can be drawn on a biological level when integrating several levels of analysis? Intuitive examples would be bulk and single-cell RNA sequencing; or is there a possibility to trace the molecular cascade all the way from phosphorylation to the proteome? Such an integration would tremendously enhance the impact of the paper.

RE: We have addressed this point in detail in our general response to all reviewers (page 1-2).

2.2. The approach to integrating mice of different sexes seems haphazard and inconsistent. The authors should explain their approach to integrating sex as a biological variable and the implications of their approach. Most experiments are done exclusively in males with the exception of a bulk RNA-sequencing experiment in males and females reported in the supplemental and a subset of TRAP-seq

experiments conducted exclusively in females. The lack of differences in bulk-RNA seq is an interesting result to report (although I note there are small and unequal sample sizes that may limit the ability to detect significant differences). However, it is not appropriate to extrapolate from this a general lack of sex differences across other untested levels of analysis. Given that an overarching conclusion of this work appears to be that the phospho-proteome is most relevant to understanding the acute stress response, it would have been interesting to see the sex comparison in this experiment. Given the inconsistencies in sex, the authors should pay attention to reporting the sex of the animals for each experiment (e.g., proteome). The title should be amended to indicate that this is in the male mouse.

RE: In response to the reviewer's comment we decided to extend our work and explore sex differences more systematically with two new approaches.

- 1) We performed a large new phosphoproteomics experiment with 12 males and 12 females evenly split into swim and control groups, to assess the phosphoproteome immediately after stress. Similarly to the RNAseq data, we found that also on the phosphoproteomic level females reacted again quite similarly (Figure 1G,H,K). Overall, however, the stress-induced changes were quantitatively larger in females (Figure 1J). For a selected number of phosphopeptides we also observed striking differences between the two sexes, which shows that at the proteomic level there are indeed robust sex effects.
- 2) We extended the analysis of sex effects on the transcriptome level. To this end we pooled data from the "timeseries experiment" and the "male vs female experiment" to specifically re-assess sex effects with an interactive sex*AS model with enough replicates (6x6 for females, 9x9 for males). Such a large N is rarely used in transcriptomic studies and is certainly unique in stress research. Overall we reproduce our earlier findings that the stress-induced changes are very similar in both sexes, and that clear sex differences are restricted to sex-chromosome-linked genes. That being said, we find indications that there are some differences between the sexes. Although the logFCs of males and females were much more aligned than in the phospho data (Figure 2H,I), it appears that also with transcriptomics several stress-induced genes tend to be more strongly regulated in females than in males. However, resolving these small differences with sufficient power would require more detailed follow-up work. We updated the results section with the new analyses and moved sex effects from the supplementary data into the main manuscript (Figure 2).

It was beyond our capability to conduct the single-nucleus RNAseq experiment in both sexes, but given that the sex differences at the bulk-mRNA level are very modest at best, we do not believe that these analyses would have further improved the manuscript. Cell-type-specific TRAP experiments were conducted in males, the ubiquitous-TRAP experiment was conducted only in females because of breeding issues with males. However, the fact that the TRAP data produce a very consistent overall picture when male and female data are pooled (see e.g. Figure 6I), we believe that the mix of male and female data here is a strength rather than a weakness and further supports the notion that sex differences are modest on the level of transcription and translation. We now briefly address these findings in the discussion (first paragraph) and we believe these additions have strengthened our manuscript considerably.

2.3. The detailed behavioral analysis is methodologically sound but feels out of context. The in-depth display of specific behavioral effects of forced swim stress do not add to the multi-omic data, especially because any specific behavior of this separate cohort of animals cannot be directly related to any of the molecular findings. Ultimately, the machine learning analysis results do not alter the interpretations based on standard behavioral testing. Further, the most interesting time points to apply a sensitive behavioral assessment which could be expected to reveal different findings (i.e. intermediate time points) are not included. I would recommend significantly shortening this part or moving substantial portions to the supplementary material.

RE: All reviewers had criticised that the in-depth, unsupervised behavior analysis added little to the main message of the paper and distracted from the molecular focus of our work. We agree with this assessment and we followed the suggestion to significantly shorten the behavior part. We thus removed the unsupervised analysis including clustering from the manuscript and moved the

remaining behavior analyses - which show that most of the stress-induced behavioral changes in the open field and elevated plus maze are very short-lived - into the supplementary section (see Figure S1). Instead, we expanded our molecular analyses - based on many reviewers' comments - which resulted in an additional main figure for snRNA data (now Figures 4 and 5).

2.4. Multiple testing corrections in the behavioral data are not reported. When comparing behaviors (here clusters) between the control and stressed group, the multiple testing correction should take into account the number of comparisons made. It is unclear whether the p-values were adjusted.

RE: Since the unsupervised behavior analysis has been removed, this issue is no longer relevant. But to answer the reviewer's question, we deliberately did not apply multiple testing corrections in this case, even though at the 45 min time point many clusters would have passed it. We did this so that we would not miss any long-lasting behavioral effects of stress exposure. The fact that only a single cluster remained significant after 24hrs - despite this lenient statistical approach - indeed suggests that most stress-induced behavior effects have disappeared at that time point.

2.5. The claim that 'no lasting alterations in anxiety-related behavior' are observed is overstated and not fully supported by the data. Both the conventional and ML analysis reveal effects at 24h post stress: increased grooming and decreased supported rearing, both of which may be interpreted as indicative of a lasting alteration in anxiety-like behavior. While this could be seen as a minor point given the lack of differences in other behavioral measures, it is worth noting given that it is at odds with the tight temporally-resolved molecular cascades reported here. The claims to complete lack of differences should be tempered and some brief discussion would be of interest.

RE: The reviewer is correct, and we have toned down the conclusions drawn from the behavior experiments. As discussed in the previous response, we have removed some of the complex unsupervised analyses, as they added little to the overall conclusion that most of the initially very strong effects of stress on behavior in the open field and elevated plus maze rapidly fade away. We also acknowledge that some molecular/structural effects must persist, as a memory of the stressful experience is certainly maintained. However, these changes might occur at a more granular level that is difficult to detect with current multi-omic approaches. We have added this statement to the first paragraph of the discussion: *"The fact that the molecular stress response is tightly controlled and terminates efficiently is in line with the short-lived increase in anxiety levels. However, a memory trace of the stressful experience and more subtle behavioral changes likely remain, and more targeted analyses at the level of synapses or cellular engrams might reveal longer-lasting molecular changes."*

2.6. The snRNA-seq data is underutilized and the reporting of this experiment is difficult to follow. Was a statistical comparison of differential expression analysis conducted on this data? It seems that lists of DEG genes identified in bulk RNA-seq were cross-referenced to FC values from snRNA-seq to attribute DEGs to specific cell types. Why not simply analyse DE in the snRNA-seq data within each cell-type? It could then be interesting to compare these results to bulk RNA-seq to examine potential convergence.

RE: We apologize that the reporting of the snRNAseq results was not clear. Differential gene expression analysis within each cell-type was indeed performed and the top results per cell-type were already reported in Figure 5A. The full result tables are now available in the repository, and the significance in the snRNAseq is now also shown in the online Stressome App). We have clarified this in the text. In addition, we now significantly extended the snRNAseq results, as described in detail in our response to all reviewers on pages 1-2.

Minor Points

I have questions concerning a number of choices made by the authors related to the study design:

2.7. The authors chose the forced swim test (FST) as an acute stressor. The authors should clearly articulate that some findings could be specific to the FST and not necessarily apply to other stressors.

RE: We agree that it is an important addition to the Discussion. Previous studies in our lab have extensively looked at the transcriptional stress response across different acute stressors in mice (and even across other models of stress) and found that they are very similar (Floriou-Servou et al, 2021, Biol Psych; Floriou-Servou et al, 2018, Biol Psych; Roszkowski et al, 2016, Neuropharmacology; Bohacek et al, 2015, Psychoneuroendocrinology). However, we certainly do not want to make the statement that this response will directly translate to all different acute stressors. We added the following clarification to the second paragraph of the discussion: *"Although our data were exclusively obtained after forced swim stress, the transcriptional response we observe in both dHC and vHC is remarkably similar to the changes previously described after novelty stress or restraint stress (Floriou-Servou et al, 2021). Thus, although some of our effects are likely modality-specific, we expect that a significant proportion of our results will also translate to different acute stress models."*

2.8. The authors report that mice were single housed 24h prior to performing the FST. However, it is not clear why this choice was made. This is somewhat concerning, given that single housing is a stressor in itself, in particular for female mice. The authors should justify this choice (e.g., by presenting pilot data or from available literature).

RE: Chronic social isolation lasting for several days or weeks is certainly a stressor for social species like mice. Short-term single-housing for 1-2 days, however, seems to relieve stress, rather than induce it. To be clear, mice were single-housed overnight (24h) before either molecular or behavioral assessments. This has become a standard procedure in our lab, ever since we showed that the rapid, stress-induced gene expression changes (e.g. immediate early genes such as cFos or Per1) are blunted in mice that are group housed (Bohacek et al. 2015). If mice are group housed, one animal must be removed from the cage for testing/stress-exposure, which "disturbs" the remaining cagemates and increases gene expression levels in those cage mates that remain in the cage (presumably because of the "social disruption"). In the same publication we also showed that single-housing mice for 24 hrs actually reduces corticosterone levels in both sexes (compared to group housed mice), and that stress-induced changes in gene expression (cFos and Per1) can be shown more clearly than in mice that are group housed and acutely separated for testing. Therefore, we believe that in order to assess the impact of acute stress, short-term single housing for 24hrs prior to the experiment actually controls several confounding variables and reduces variability by ensuring that all mice experience the same pre-testing environment. All publications from our group have since used this protocol (e.g. Roszkowski et al, Neuropharmacology, 2016; Sturman et al, Stress 2018; Floriou-Servou et al, Biol Psych 2018; Zerbi et al, Neuron 2019; Sturman et al, Neuropsychopharmacology 2020). We have added a statement about this rationale and a reference to the methods section (section: "Animals").

2.9. It is unclear why a given series of time points was chosen for each level of analysis (e.g., transcriptomic analysis). Were there pilot studies to determine the optimal time points? Or was this based on time points reported in the literature?

RE: Whenever possible we based our choices on previous literature and we have now added the relevant references in the text. Here we provide more details about our reasoning:

Phosphoproteomics: Prior studies have shown nearly instantaneous activation of selected phosphorylation events after stress (e.g. Shen et al, *BMC Neurosci* 2004). However the literature did not provide adequate data to determine at what time an offset should be expected. Therefore we assumed that most of the changes on this level probably happen prior to the onset of the transcriptomic response (30-45 min).

RNasequencing: Here we had more information to build on. In a prior study we have analyzed a number of immediate early genes using qPCRs. (Roszkowski et al, 2016, Neuropharmacology). We found that the first gene expression changes appeared after 30 min, many genes peaked around 45 min, some at 90-120 min. Other studies sampled gene expression 1h, 3h and 5h after corticosterone

stimulation of hippocampal neurons in vitro and found transcriptomic waves at 1h and 3h, but not at 5h) (Morsink et al, 2006, J Neuroendocrinol). We therefore thought that 4 hours would be a sensible last time point. We would have added more time points if we had seen persistent gene expression changes at 4h, which was not the case.

Proteomics: Many studies have used western blotting to report single protein changes within a few hours after stress exposure, although typically this involved immediate early genes like cFos, known to peak around 1.5 - 2 hours after stress initiation. We reasoned that it would be sensible to assess proteome-wide changes at the 4h time point for several reasons: 1) We were interested to see which gene expression changes are turned into changes on the protein level, and the transcriptomic response terminated at 4h. 2) We were interested in longer-lasting protein changes that would occur after the first wave of immediate early gene expression. 3) Protein changes are usually longer lived than mRNA changes, so we reasoned that it would make sense to first allow all mRNA translation to play out.

2.10. Why cut the hippocampus into two halves rather than thirds? The referenced paper from Fanselow & Dong identifies three divisions along the dorsoventral axis: dorsal, intermediate and ventral. Could inclusion of intermediate hippocampus in dorsal and ventral contribute to observed correlations?

RE: The reviewer is correct that previous work by Fanselow and Dong (2009) and several others (Cembrowski et al, 2016, Neuron; Cembrowski et al, 2016, eLife; Floriou-Servou et al, 2018, Biol Psych) has shown a gradient of gene expression from dorsal to ventral hippocampus. Thus subdivision into 3 or more structures would have been reasonable. However, having 3 rather than 2 samples for each animal dramatically increases the large financial burden of a multi-omic approach across multiple time points. Further, our recent meta-analysis shows that - despite the extremely profound differences in baseline gene expression - the stress response on the transcriptomic level is highly conserved between dHC and vHC (Floriou-Servou, Biol Psych, 2021). Thus the response in dHC and vHC is very similar, it just starts from a different baseline. The large amount of data presented in this manuscript also strongly suggests that the stress-induced changes in dHC and vHC are highly conserved and strong differences in stress-responsiveness between these regions are the exception rather than the norm. However, we agree that separate analyses on the intermediary HC might have revealed additional nuances that may have gotten diluted due to our dissection strategy.

2.11. Some of the plots are hard to read because the authors try keeping identical axis limits across different subplots (e.g., Fig 3, B E F). While this is generally a good approach, it becomes limiting when the data points are shrunken such that differences between plots can no longer be appreciated. I would suggest flexible axis labeling in cases where it would enhance readability of the plot for the readability.

RE: Thank you for this helpful suggestion. As also recommended by reviewer 1, we opted to use ceiling values. These ceilings are now indicated in the Figure legend (now Figure 2B & Figure S3D/E).

2.12. Group ordering in some plots is not intuitive. For example 4A presents processed to the left of unprocessed. This is a minor point but reversing order such that unprocessed precedes processed would make interpretation of claims relating to temporal sequence of events easier to visually inspect in the figures

RE: We thank the reviewer for this observation. The ordering has been reversed (now Figure 3A).

2.13. For any k-means clustering, it is not reported how the number of initial centers was chosen. This can be influential on the outcome if the number of centers initially chosen is too small.

RE: Thank you for pointing out this omission. Please see our answer to reviewer #1 (point 1.2).

2.14. I would refrain from reporting on results with 50% FDR, as this indicates that half of what is reported may be false (see Results/Proteome). I would suggest removing these results from the text.

RE: We have removed the 50% FDR from the volcano plots in Figure 7E, to avoid misleading the reader. However, we believe that reporting results with 50% FDR can still be useful for generating hypotheses, especially when using techniques such as proteomics, which are difficult to implement for most labs and might also lack the sensitivity to easily detect subtle changes. As we thought that it is interesting that the dentate gyrus of the vHC seemed more responsive to stress, and that ribosomal networks seem to get engaged, we decided to keep Figure 7F. However, we adjusted the text to clearly point out the exploratory nature of this pathway analysis: *"To perform an exploratory analysis, we relaxed the significance cutoff to a FDR of 0.5, which revealed subtle changes exclusively in the DG of the vHC, and a protein-protein interaction analysis showed an enrichment for ribosomal proteins."* (page 10 and Figure 7F)

2.15. The authors should limit descriptions to observable behavioral states. For example, rather than describing mice as 'anxious', discuss the relevant behavioral metrics that indicate anxiety-like behavior.

RE: We have adjusted the behavior section accordingly, we now name the observed behaviors directly, and refer to the general stress-induced changes in behavior as "favoring avoidance over exploration" (see "Results" section, first paragraph).

2.16. Reporting of results would benefit from more specific conclusions. For example, a detailed analysis of SYN1 phosphorylation is presented but it is unclear what we should conclude from the findings.

RE: Throughout the results section we tried to add conclusions whenever possible. We added a sentence to better clarify that the goal of the more detailed Syn1 phosphorylation analysis was to demonstrate the depth quality of the phosphoproteomic data that we make accessible for researchers (page 5, first paragraph). Syn1 is a target of the MAPK/ERK-CREB pathway and one of the few proteins for which we found published data on the phosphoproteomic level after stress (Revest et al., 2010), however their findings lacked site specificity. Using our data, we can show that not only can we replicate these published results, but we can further pin-point the exact sites of the phosphorylation changes. Since this manuscript is intended as a resource paper, we believe this example demonstrates the usefulness for other research groups.

Reviewer #3 (Remarks to the Author):

von Ziegler and colleagues examined the molecular cascades that occur in mouse hippocampus following a brief forced swim stressor. The authors broadly found that gene expression, protein phosphorylation, and protein translation events that occur in response to acute stress resolve within four hours of the stressor, which parallels the brief timescale of behavioral response to the stressor. The authors describe this acute stress response as adaptive homeostasis. This study is particularly valuable as the authors investigate multiomic molecular response across a time course, and because of the interactive web portal authors have developed to public browsing of their datasets. There is a minor attempt to understand sex differences in acute stress response. Analysis of hippocampal subregion specificity is useful given previous evidence on separate dorsal/ventral and CA1/3/DG functions in stress response, and it is also of interest that the authors discovered correlation between expression changes at early time points, indicating similarity in early stress response. The analyses examining correlations of change between consecutive time points to discover that expression/protein changes are not random but unfold over time are particularly elegant. Cell-type specificity of transcriptomic change at the 45m time point is also made, in male (?) whole-hippocampus. This could easily be several papers, but is nevertheless useful as one complete piece. Overall, the science is extremely well-done and this manuscript is well-organized and well-written. I have only minor comments to clarify points for this exciting body of work.

RE: We thank the reviewer for the very positive evaluation of our work and for the constructive feedback.

Minor Comments:

1. The overall conclusions from the DLC-based behavioral characterization presented in Figure 1 are very useful, although the conclusions are not different from those garnered from the gross-scale behavioral characterization (S1). The main difference is that the DLC-based analysis is very complicated and non-intuitive. In particular, the correlation analysis does not lead to any conclusions useful for understanding resolution of the stress response. This figure would be more useful if the authors overlaid information given in the text on specific types of behaviors post-hoc associated with the behavior clusters (rears, walking, grooming). Perhaps a more useful correlation would be a 4h vs 24h comparison.

Re: All reviewers had criticised that the in-depth, unsupervised behavior analysis added little to the main message of the paper and distracted from the molecular focus of our work. We agree with this assessment and we followed the suggestion of Reviewer 2 to significantly shorten the behavior part. We thus removed the unsupervised analysis including clustering from the manuscript and moved the remaining behavior analyses - which show that most of the stress-induced behavioral changes in the open field and elevated plus maze are very short-lasting - into the supplementary section (Figure S1).

2. It is unclear in many of the heatmaps how the data is sorted along the y-axes (the types and time bins make sense). For example, in 4A and 4D, it appears vaguely but not-quite sorted by expression change at 45 min. Sorting in 5D-E are also unclear. Can the authors clarify how genes are ordered in the figure legends?

RE: The rows were sorted in an unsupervised fashion, using the genes' angle on a two-dimensional projection of z-scores. We found this approach, which is generally less fragile to variations in the data and in parameters than hierarchical clustering, to provide the most intelligible overview especially for temporal responses. We now clarified this in the methods "*Unless otherwise specified, the rows were sorted using the features' angle on a two-dimensional projection of the plotted values, as implemented in STools.*" (page 17).

3. It is interesting that the snRNA-seq generally did not show expression differences between Ctl and AS samples within any cell type. The authors have done a nice job using that data to understand the bulk RNA-seq expression differences, but additional discussion of the lack of snRNAseq differences in warranted.

RE: We apologize that the reporting of the snRNAseq results was not clear, all reviewers struggled with this section. We did in fact find a substantial number of differentially expressed genes in the various cell types of the snRNAseq, and the top ones were also reported (Figure 5A). However, we focused more on the integration than on the snRNAseq on its own, partly because the bulk data has many more replicates than the snRNAseq, and partly in an effort to offer a broader picture of the transcriptional response. We have now clarified this in the text, and in response to other reviewers' comments we have further developed the snRNAseq analysis and updated the Stressome APP so it also reports statistical significance for individual genes in the snRNAseq data.

4. Why would statistical power of TRAP-seq be lower than bulk-RNAseq, given the higher specificity of TRAP-seq for active changes? The authors have attempted to get around a lack of significant alterations by re-combining with bulk-seq data, but the justification for this is insufficient. The rest of the manuscript is so thorough, it does not need the less-conclusive TRAP data, despite all of the work that surely went into these experiments. The cellular specificity of some of the IEGs is interesting, but surely the snRNAseq data should point to the same conclusions?

RE: Due both to the highly dynamic nature of the transcriptional response and the specificities of the two protocols, they provide very different information (see response 1.4 to reviewer #1). For example, the genes shown to be cell-type-specific in the TRAP cannot be directly found in the snRNAseq because their increase in transcription is prior to the 45 min time point. In addition, our independent analyses restricted to individual data-sets (independent TRAP analyses of the Camk2a, Viat and CMV cohorts, swim vs. control groups) do show very significant genes (Figure S6B/C), even

though they are few in number. These results are consistent with our recently conducted re-analysis of published TRAP data from various acute stress experiments in the McEwen lab (von Ziegler et al, Biorxiv, 2020; <https://www.biorxiv.org/content/10.1101/2020.11.24.392464v1>). Further, they recapitulate, albeit at a much weaker magnitude, the genes most strongly increased in translation in a published TRAP after 1h sustained KA exposure (Fernandez-Albert et al., 2019; see Figure S7E). We are thus confident that we can accurately identify translational changes using our TRAP approach. Nevertheless, we do expect a lower statistical power in TRAP due to its higher variability compared to bulk sequencing, as we show in Figure S6D. Unfortunately, this is intrinsic to the method that uses many technical steps during pull-down that can be somewhat variable between samples. That being said, the variability of our TRAP dataset was considerably better (lower) than that of published datasets (e.g. Marrocco et al. 2019) (Figure S6D), which we recently re-analyzed (von Ziegler et al, 2020, BioRxiv). Despite the reduced statistical power, which we tried to mitigate with an increased sample size, the various lines of evidence (including also the proteomics data) converge towards the important hypothesis of translational buffering. This is corroborated by the observation that, even in the very strong response in Camk2a-TRAP elicited by sustained kainate (Fernandez-Albert et al., 2019) (Figure S7E), a non-linearity can be observed in the transcriptome-translatome comparison, with lower RNA fold-changes being partially buffered. We now address this more thoroughly in the discussion (page 12). To summarize, while the limitations of TRAP certainly reduce our power to detect small changes, the translational response is much milder and selective than the transcriptional one, and we believe this data is critical to the understanding of the stress response.

Reviewers' Comments:

Reviewer #2:

Remarks to the Author:

The authors are to be commended for this incredibly thorough, constructive and thoughtful response addressing the issues raised in the first round of reviews. They have comprehensively integrated the comments from all three reviewers, adding substantial new data and analyses as well as important clarifications and text edits to return a greatly strengthened paper that will undoubtedly make an important contribution to the field. I am satisfied that the manuscript is suitable for publication.

Reviewer #3:

Remarks to the Author:

The authors have done a remarkable job with this complex dataset and have been responsive to reviewer critiques. The datasets are open (through the interactive app built) and other researchers can validate or extend the analysis. For these reasons, the manuscript is ready for publication in my opinion.